# Understanding Mental Representations Of Objects Through Verbs Applied To Them

## Abstract

In order to interact with objects in our environment, we rely on an understanding of the actions that can be performed on them, and the extent to which they rely or have an effect on the properties of the object. This knowledge is called the object "affordance". We propose an approach for creating an embedding of objects in an affordance space, in which each dimension corresponds to an aspect of meaning shared by many actions, using text corpora. This embedding makes it possible to predict which verbs will be applicable to a given object, as captured in human judgments of affordance, better than a variety of alternative approaches. Furthermore, we show that the dimensions learned are interpretable, and that they correspond to typical patterns of interaction with objects. Finally, we show that the dimensions can be used to predict a state-of-the-art mental representation of objects, derived purely from human judgements of object similarity.

## 1 Introduction

In order to interact with objects in our environment, we rely on an understanding of the actions that can be performed on them, and their dependence (or effect) on properties of the object. Gibson (1979) coined the term "affordance" to describe what the environment "provides or furnishes the animal". Norman (2013) developed the term to focus on the properties of objects that determine the action possibilities. The notion of "affordance" emerges from the relationship between the properties of objects and human actions. If we consider "object" as meaning anything concrete that one might interact with in the environment, there will be thousands of possibilities, both animate and inanimate (see WordNet (Miller, 1998)). The same is true if we consider "action" as meaning any verb that might be applied to the noun naming an object (see VerbNet (Schuler, 2005)). Intuitively, only a relatively small fraction of all possible combinations of object and action will be plausible. Of those, many will also be trivial, e.g. "see" or "have" may apply to almost every object. Finally, different actions might reflect a similar mode of interaction, depending on the type of object they are applied to (e.g. "chop" and "slice" are distinct actions, but they are both used in food preparation).

Mental representations of objects encompass many aspects beyond function. Several studies (McRae et al., 2005; Devereux et al., 2014; Hovhannisyan et al., 2020) have asked human subjects to list binary properties for hundreds of objects, yielding thousands of answers. Properties could be taxonomic (category), functional (purpose), encyclopedic (attributes), or visual-perceptual (appearance), among other groups. While some properties were affordances in themselves (e.g. "edible"), others reflected many affordances at once (e.g. "is a vegetable" means that it could be planted, cooked, sliced, etc). More recently, Zheng et al. (2019); Hebart et al. (2020) introduced SPoSE, a model of the mental representations of objects. The model was derived from a dataset of 1.5M Amazon Mechanical Turk (AMT) judgments of object similarity, where subjects were asked which of a random triplet of objects was the odd one out. The model was an embedding for objects where each dimension was constrained to be sparse and positive, and where triplet judgments were predicted as a function of the similarity between embedding vectors of the three objects considered. The authors showed that these dimensions were predictable as a *combination* of elementary properties in the Devereux et al. (2014) norm that often co-occur across many objects. Hebart et al. (2020) further showed that 1) human subjects could coherently label what the dimensions were "about", ranging from categorical (e.g. is animate, food, drink, building) to functional (e.g. container, tool) or structural (e.g. made of metal or wood, has inner structure). Subjects could also predict what dimension values new objects would

have, based on knowing the dimension value for a few other objects. SPoSE is unusual in its wide coverage – 1,854 objects – and in having been validated in independent behavioral data.

Our first goal is to produce an analogous affordance embedding space for objects, where each dimension groups together actions corresponding to a particular "mode of interaction"affordance mining. Our second goal is to understand the degree to which affordance knowledge underlies the mental representation of objects, as instantiated in SPoSE. In this paper, we will introduce and evaluate an approach for achieving both of these goals. Our approach is based on the hypothesis that, if a set of verbs apply to the same objects, they apply for similar reasons. We start by identifying applications of action verbs to nouns naming objects, in large text corpora. We then use the resulting dataset to produce an embedding that represents each object as a vector in a low-dimensional space, where each dimension groups verbs that tend to be applied to similar objects. We do this for larger lists of objects and action verbs than previous studies (thousands in each case). Combining the weights on each verb assigned by various dimensions yields a ranking over verbs for each concept. We show that this allows us to predict which verbs will be applicable to a given object, as captured in human judgments of affordance. Further, we show that the dimensions learned are interpretable, and they group together verbs that would all typically occur during certain complex interactions with objects. Finally, we show that they can be used to predict most dimensions of the SPoSE representation, in particular those that are categorical or functional. This suggests that affordance knowledge underlies much of the mental representation of objects, in particular semantic categorization.

## 2 RELATED WORK

The problem of determining, given an action and an object, whether the action can apply to the object was defined as "affordance mining" in Chao et al. (2015). The authors proposed complementary methods for solving the affordance mining problem by predicting a plausibility score for each combination of object and action. The best method used word co-occurrences in two ways: n-gram counts of verb-noun pairs, or similarity between verb and noun vectors in Latent Semantic Analysis (Deerwester et al., 1990) or Word2Vec (Mikolov et al., 2013) word embeddings. For evaluation, they collected AMT judgements of plausibility ("is it possible to $< verb >$ a $< object >$") for every combination of 91 objects and 957 action verbs. The authors found they could retrieve a small number of affordances for each item, but precision dropped quickly with a higher recall. Subsequent work (Rubinstein et al., 2015; Lucy & Gauthier, 2017; Utsumi, 2020) predicted properties of objects in the norms above from word embeddings (Mikolov et al., 2013; Pennington et al., 2014), albeit without a focus on affordances. Forbes et al. (2019) extracted 50 properties (some were affordances) from Devereux et al. (2014), for a set of 514 objects, to generate positive and negative examples for 25,700 combinations. They used this data to train a small neural network to predict these properties. The input to the network was either the product of the vectors for object and property, if using word embeddings (Mikolov et al., 2013; Pennington et al., 2014; Levy & Goldberg, 2014), or the representation of a synthesized sentence combining them, if using contextualized embeddings (Peters et al., 2018; Devlin et al., 2018). They found that the latter outperformed the former for property prediction, but none allowed reliable affordance prediction. In addition to object/action plausibility prediction, Ji et al. (2020) addressed the problem of determining whether a object1/action/object2 (target of the action with object1) was feasible. They selected a set of 20 actions from Chao et al. (2015) and combined them with the 70 most frequent objects in ConceptNet (Speer & Havasi, 2012) into 1400 object/action pairs, which were then labelled as plausible or not; given rater disagreements, this yielded 330 positive pairs and 1070 negative ones. They then combined the positive pairs with other objects as "tails" (recipients of the action), yielding 3900 triplets. They reached F1 scores of 0.81 and 0.52 on the two problems, respectively. Other papers focus on understanding the relevant visual features in objects that predict affordances, e.g. (Myers et al., 2015; Sawatzky et al., 2017; Wang & Tarr, 2020). The latter collected affordance judgments on AMT ("what can you do with $< object >$") for 500 objects and harmonized them with WordNet synsets for 334 action verbs. For validation of the rankings of verb applicability predicted by our model, we will use the datasets from Chao et al. (2015) and Wang & Tarr (2020), as they are the largest available human rated datasets.

In robotics research, affordance refer to relation between agent, action and the environment, under the constraints of motor and sensing capabilities of the agent (Lopes et al., 2007). Affordance modeling for robotics have been studied extensively, see recent surveys (Jamone et al., 2016; Zech et al., 2017; Hassanin et al., 2018). Due to the restriction in action possibilities and the complexity of

real world scenarios, poor semantic generalization of affordance inference is observed in visual and speech-based robotic systems. Attempts to use semantic role labelling or text corpora, in tandem with visual features, have been proposed (e.g Persiani & Hellström (2019); Chen et al. (2020). In general, though, this work focuses on a restricted set of objects and manipulation actions.

In computational linguistics, (Resnik, 1993) introduced computational approaches to determining selectional preference, the degree to which a particular semantic class tends to be used as an argument to a given predicate. From this perspective, the method we introduce in this paper provides a way of scoring verbs for how well they would apply to objects. All methods proposed to do this take advantage of some grouping of verbs and objects into classes (e.g. WordNet in Resnik (1996), or co-occurrence statistics of words in a corpus (Erk, 2007; Padó et al., 2007; Séaghdha, 2010; Van De Cruys, 2014; Zhang et al., 2020). Erk (2007) used similarities of co-occurrence patterns for words to compute selectional preferences for semantic roles in FrameNet. Padó et al. (2007) used the same similarity function to predict the plausibility of verb/relation/argument triples. Séaghdha (2010) learned a custom topic model with separate latent variables (and dictionaries) for verbs and nouns. Van De Cruys (2014) trained a neural network to predict preference scores for combinations of verbs or objects, represented via embedding vectors. Zhang et al. (2020) learns embeddings for individual words together with modifications for when the word is used in a certain relation. It scores combinations of words by similarity of the modified embedding vectors. All of these methods could be used to score verbs by how applicable they are to a given noun, the ancillary task we use to make sure our proposed embedding carries the relevant information. That said, they would require substantial implementation effort, even where the code is publicly available. This said, making that prediction is not our main goal. Our proposed embedding space is a latent variable model for verb-noun applications. While this is also the case for the dual topic model in (Séaghdha, 2010) the internal representation in Van De Cruys (2014), or the embeddings of nouns/verbs in Zhang et al. (2020), they would all require extensive modification to add sparsity assumptions – important for interpretability – and to produce combined verb rankings for embedding vectors.

## 3 DATA AND METHODS

### 3.1 OBJECTS AND ACTIONS CONSIDERED

In this paper, we use the list of 1854 object concepts introduced in (Hebart et al., 2019) and for which SPoSE embeddings are available. This list sampled from concrete, picturable, and nameable nouns in American English, and was further expanded by a crowdsourced Amazon Mechanical Turk study to elicit category membership ratings. The following 27 categories accounted for most of the objects: food, animal, clothing, tool, drink, vehicle, fruit, vegetable, body part, toy, container, bird, furniture, sports equipment, musical instrument, dessert, part of car, weapon, plant, insect, kitchen tool, office supply, clothing accessory, kitchen appliance, home decor, medical equipment, and electronic device. As we are not doing sense disambiguation for each noun that names an object, we will use noun or object interchangeably throughout the paper. We created our own verb list by having three annotators go through all verb categories on VerbNet (Schuler, 2005), and selecting those that included verbs that corresponded to an action performed by a human on an object. We kept all the verbs in each selected category, and selected only those categories where all annotators agreed. Those VerbNet categories contained $\sim 10-50$ verbs sharing thematic roles and selectional preferences (e.g. fill-9.8, amalgamate-22.2, manner-speaking-37.3, build-26.1, remove-10.1, cooking-45.3, create-26.4, destroy-44, mix-22.1, vehicle-51.4.1, dress-41.1.1). The resulting list has 2541 verbs.

### 3.2 EXTRACTION OF VERB APPLICATIONS TO NOUNS FROM TEXT CORPORA

We used the UKWaC and Wackypedia corpora (Ferraresi et al.) containing, approximately, 2B and 1B tokens, and 88M and 43M sentences, respectively. The former is the result of a crawl of British web pages, while the latter is a subset of Wikipedia. Both have been extensively cleaned up and have clearly demarcated sentences, which makes them ideal for dependency parsing. We replaced all common bigrams in Brysbaert et al. (2014) by a single token. We identified all sentences containing both verbs and nouns in our list, and we used Stanza (Qi et al., 2020) to produce dependency parses for them. We extracted all the noun-verb pairs in which the verb was a syntactic head of a noun having `obj` (object) or `nsubj:pass` (passive nominal subject) dependency relations. We compiled raw counts of how often each verb was used on each noun, producing a count matrix $M$. Note that

this is very different from normal co-occurrence counts - those would register a count whenever verb and noun were both present within a short window (e.g. up to 5 words away from each other), *regardless* of whether the verb applied to the noun, or they were simply in the same sentence. Out of the 1854 nouns considered, there were 1755 with at least one associated action, and this is the list we will use in the remainder of this paper. Note also that the counts pertain to every possible meaning of the noun, given that no word sense disambiguation was performed.

Finally, we converted the matrix $M$ into a Positive Pointwise Mutual Information (PPMI (Turney & Pantel, 2010)) matrix $P$ where, for each object $i$ and verb $k$:

$$P(i,k) := \max\left(\log\frac{\mathbb{P}(M_{ik})}{\mathbb{P}(M_{i*}) \cdot \mathbb{P}(M_{*k})}, 0\right), \tag{1}$$

where $\mathbb{P}(M_{i*})$ and $\mathbb{P}(M_{*k})$ are respectively the marginal probability of $i$ and $k$. $P$ can be viewed as a pair-pattern matrix (Lin & Pantel, 2001), where the PPMI helps in separating frequency from informativess of the co-occurrence of nouns and verbs (Turney & Pantel, 2010; Turney & Littman, 2003). However, PPMI is also biased and may be large for rare co-occurrences, e.g. for $(o_i, v_k)$ that co-occur only once in $M$. This is addressed in the process described in the next section.

### 3.3 OBJECT EMBEDDING IN A VERB USAGE SPACE

**Object embedding via matrix factorization**   Our embedding is based on a factorization of the PPMI matrix $P$ ($m$ objects by $n$ verbs) into the product of low-rank matrices $O$ ($m$ objects by $d$ dimensions) and $V$ ($n$ verbs by $d$ dimensions), yielding $\widetilde{P} := OV^T \approx P$. $O$ is the object embedding in $d$-dimensional space, and $V$ is the weighting placed on each verb by each dimension.

Intuitively, if two verbs occur often with the same objects, they will both have high loadings on one of the $d$-dimensions; conversely, the objects they occur with will share high loadings on that dimension. The idea of factoring a count matrix (or a transformation of it) dates back to Latent Semantic Analysis (Landauer & Dumais, 1997), and was investigated by many others ((Turney & Pantel, 2010) is a good review). Given that PPMI is biased towards rare pairs of noun/verb, the matrix $P$ is not necessarily very sparse. If factorized into a product of two low-rank matrices, however, the structure of the matrix can be approximated while excluding noise or rare events (Bullinaria & Levy, 2012).

**Optimization problem**   Given that PPMI is positive, the matrices $O$ and $V$ are as well. This allows us to obtain them through a non-negative matrix factorization (NMF) problem

$$O^*, V^* = \underset{O,V}{\operatorname{argmin}} \|P - OV^T\|_F^2 + \beta\mathcal{R}(O,V), \tag{2}$$

which can be solved through an iterative minimization procedure. For the regularization $\mathcal{R}(O,V)$, we chose the sparsity control $\mathcal{R}(O,V) \equiv \sum_{ij} O_{ij} + \sum_{ij} V_{ij}$.

In our experiments, we use $d = 70$ and $\beta = 0.3$. These values were found using the two-dimensional hold-out cross validation (Kanagal & Sindhwani, 2010), due to its scalability and natural fit to the multiplicative update algorithm for solving (2). Specifically, denoting $M_t$ and $M_v$ to be the mask matrices representing the held-in and held-out entries, we optimize for

$$O^*, V^* = \underset{O,V}{\operatorname{argmin}} \|M_t \odot (P - OV^T)\|_F^2 + \beta\mathcal{R}(O,V) \tag{3}$$

and obtain the reconstruction error $E = \|M_v \odot (P - O^*(V^*)^T)\|_F^2 + \beta\mathcal{R}(O^*, V^*)$. For more details, please refer to Appendix A. The optimization problem (2) is NP-hard and all state-of-the-art algorithms may converge only to a local minimum (Gillis, 2014); choosing a proper initialization of $O$ and $V$ is crucial. We used the NNDSVD initialization (Boutsidis & Gallopoulos, 2008), a SVD-based initialization which favours sparsity on $O$ and $V$ and approximation error reduction.

**Estimating the verb usage pattern for each object**   Each column $V_{:,k}$ of matrix $V$ contains a pattern of verb usage for *dimension $k$*, which captures verb co-occurrence across all objects. Deriving a similar pattern for each *object $i$*, given its embedding vector $O_{i,:} = [o_{i_1}, o_{i_2}, \ldots o_{i_d}]$, requires combining these patterns based on the weights given to each dimension. The first step in doing so

requires computing the cosine similarity between each embedding dimension $O_{:,h}$ and the PPMI values $\tilde{P}_{:,k}$ for each verb $k$ in the approximated PPMI matrix $\tilde{P} = OV^T$, which is

$$S(O_{:,h}, \tilde{P}_{:,k}) = \frac{O_{:,h} \cdot \tilde{P}_{:,k}}{\|O_{:,h}\|_2 \|\tilde{P}_{:,k}\|_2}. \tag{4}$$

Given the embedding vector for object $i$, $O_{i,:} = [o_{i_1}, o_{i_2}, \ldots o_{i_d}]$, we compute the pattern of verb usage for the object as $O_{i,:}S$. Thus, this is a weighted sum of the similarity between every $O_{:,h}$ and $\tilde{P}_{:,k}$. We will refer to the ordering of verbs by this score as the *verb ranking* for object $i$.

## 4 EXPERIMENTS AND RESULTS

### 4.1 PREDICTION OF AFFORDANCE PLAUSIBILITY

**Affordance ranking task**    The first quantitative evaluation of our embedding focuses on the ranking of verbs as possible affordances for each object. We will use the Affordance Area Under The Curve (AAUC) relative to datasets that provide, for each object, a set of verbs known (or likely) to be affordances. Intuitively, the verb ranking for object $i$ is good if it places these verbs close to the top of the ranking, yielding an AAUC of 1. Conversely, a random verb ranking would have an AAUC of 0.5, on average. Note that this is a conservative measure, given that a perfect ranking would still penalize every true affordance not at the top. Hence, this is useful as a *relative* measure for comparing between our and competing approaches for producing rankings. More formally, given the $K$ ground truth verb affordances $\{g_k\}_{k=1}^K$ of object $i$, and its verb ranking $\{v_i\}_{i=1}^n$, we denote $\ell_k$ to be the index such that $v_{\ell_k} = g_k \ \forall k$. We then define AUCC for object $i$ as AUCC $= \frac{1}{K} \sum_{k=1}^K \left(1 - \frac{\ell_k}{n}\right)$.

**Datasets**    We used the two largest publicly available object affordance datasets as ground truth. In the first dataset, WTAction (Wang & Tarr, 2020), objects are associated with the top 5 actions label provided by human annotators in response to "What can you do with this object?". Out of 1,046 objects and 445 actions in this dataset, there are 971 objects and 433 verbs that overlap with those in our lists ($\sim 3.12$ action labels per object) . The second dataset, MSCOCO (Chao et al., 2015), scores every candidate action for an object ranging from 5.0 ("definitely an affordance") to 1.0 ("definitely not an affordance") marked by 5 different workers. We consider only a 5.0 score as being an affordance, whereas Chao et al. (2015) used both 4.0 and 5.0. Out of 91 objects and 567 actions, there are 78 objects and 558 verbs that overlap with ours ($\sim 34$ action labels per object).

**Baseline methods**    We compared the ranking of verbs produced by our algorithm with an alternative proposed in (Chao et al., 2015): ranking by the cosine similarity between word embedding vectors for each noun and those for all possible verbs in the dataset. We considered several off-the-shelf embedding alternatives, namely Word2Vec (Mikolov et al. (2013), 6B token corpus), GloVe (Pennington et al. (2014), 6B and 840B token corpora, and our 2B corpus), Dependency-Based Word Embedding (DBWE, Levy & Goldberg (2014), 6B corpus), and Non-negative Sparse Embedding (NNSE,Murphy et al. (2012), 16B corpus). Finally, we included the other two methods in (Chao et al., 2015), LSA (Deerwester et al. (1990), trained on our corpus), and ranking by frequency of verb/noun pair in Google N-grams (Lin et al. (2012)). The embeddings are 300-D in all cases, except for NNSE (1000-D, results are similar for 2500-D). We contrasted Word2Vec and GloVe because they are based on two different embedding approaches (negative sampling and decomposition of a word co-occurrence matrix), developed on corpora twice as large as ours. We contrasted 6B and 840B versions of GloVe to see the effect of increasing dataset size, and 2B to show the effect of our corpus. In these methods, the co-occurrence considered is simply proximity within a window of a few tokens, rather than application of the verb to the noun. We included DBWE because it uses dependency parse information (albeit to define the word co-occurrence window, rather than select verb applications to nouns as we do). We included NNSE because it is based on a sparse, non-negative factorization of a co-occurrence matrix of features derived from words combined with specific dependency relations. Finally, we ranked the verbs by their values in the row of the PPMI matrix $P$ corresponding to each probed object, to see the effect of using a low-rank approximation in extracting information.

**Results**    For each dataset, we reduced our embeddings $O$ and $V$ according to the sets of objects and verbs available. We then obtained the verb ranking for each object, as described in Section 3.3,

Table 1: Average AAUC of verb rankings produced by our and baseline methods

| Dataset | | DBWE | NNSE (1000-D) | N-gram | W2V (6B) | GloVe (2B) | GloVe (6B) | GloVe (840B) | LSA | PPMI | Ours |
|---------|------|------|------|------|------|------|------|------|------|------|------|
| WTaction | $\mu$ | 0.60 | 0.65 | 0.60 | 0.70 | 0.79 | 0.75 | 0.80 | 0.81 | 0.77 | **0.88** |
| | $S_E$ | 6e−3 | 8e−3 | 8e−3 | 7e−3 | 7e−3 | 7e−3 | 6e−3 | 5e−3 | 6e−3 | **4e−3** |
| MSCOCO | $\mu$ | 0.56 | 0.58 | 0.53 | 0.59 | 0.66 | 0.64 | 0.68 | 0.63 | 0.61 | **0.77** |
| | $S_E$ | 8e−3 | 9e−3 | 8e−3 | 7e−3 | 7e−3 | 8e−3 | 6e−3 | 7e−3 | 8e−3 | **5e−3** |

as well as rankings predicted with the different baseline methods in the previous section. Table 1 shows the AAUC results (in terms of the mean $\mu$ and the standard error $S_E$ of AAUCs) obtained with these verb rankings on the two datasets. Our ranking is better than those of all the baseline methods, as determined from a paired two-sided $t$-test, in both WTAction ($p$-values of 7.36e−23, 4.19e−14, 6.82e−174, 1.34e−53, 1.44e−89, 5.21e−47, 8.70e−25 and 8.14e−39) and MSCOCO ($p$-values of 1.54e−23, 3.37e−21, 9.32e−36, 2.15e−27, 2.82e−30, 9, 64e−20, 7.93e−17 and 2.13e−29).

The overall distributions of AAUCs for the two datasets are shown in Appendix B. Our procedure yields AAUC closer to 1.0 for many more items than the other methods. This suggests that the co-occurrence of verb and noun within a window of a few tokens, the basis of the word embeddings that we compare against, carries some information about affordances, but also includes other relationships and noise (e.g. if the verb is in one clause and the noun in another). Results are better in the embedding trained in a corpus 280X larger, but still statistically worse than those of our procedure. Ranking based on the PPMI matrix $P$ performs at the level of the 6B token embeddings. This suggests that our procedure is effective at removing extraneous information in the matrix $P$.

## 4.2 PREDICTION OF SPoSE OBJECT REPRESENTATIONS FROM OUR AFFORDANCE EMBEDDING

**The SPoSE representation and dataset** The dimensions in the SPoSE representation (Hebart et al., 2020) are interpretable, in that human subjects coherently label what those dimensions are "about", ranging from the categorical (e.g. animate, building) to the functional (e.g. can tie, can contain, flammable) or structural (e.g. made of metal or wood, has inner structure). Furthermore, subjects could also predict what dimension values new objects would have, based on knowing the dimension value for a few other objects. The SPoSE vectors for objects are derived solely from behaviour in a "which of a random triplet of objects is the odd one out" task. The authors propose a hypothesis for why there is enough information in this data to allow this: when given any two objects to consider, subjects mentally sample the contexts where they might be found or used. The resulting dimensions reflect the aspects of the mental representation of an object that come up in that sampling process. The natural question is, then, which of these dimensions reflect affordance or interaction information, and that is what our experiments aim to answer.

For our experiments, we used the 49-D SPoSE embedding published with (Hebart et al., 2020). Out of these, we excluded objects named by nouns that had no verb co-occurrences in our dataset and, conversely, verbs that had no interaction with any objects. We averaged the vectors for each set of objects named by the same polysemous noun (e.g. "bat"). This resulted in a dataset of 1755 objects/nouns, with their respective SPoSE embedding vectors, and 2462 verbs.

**Prediction of SPoSE meaning dimensions from affordance embedding** The first experiment we carried out was to predict the SPoSE dimensions for an object from its representation in our affordance embedding. The ability to do this tells us which SPoSE dimensions – and, crucially, which *kinds* of SPoSE dimensions – can be explained in terms of affordances, to some extent.

Denoting the SPoSE embeddings for $m$ objects as a $m \times 49$ matrix Y, we solved the following Lasso regression problem for each column $Y_{:,i}$

$$w_i^* = \arg \min_{w \in \mathbb{R}^d, w \geq 0} \frac{1}{2m} \|Y_{:,i} - Ow\|_2^2 + \lambda \|w\|_1, \quad i = 1, 2, \ldots 49, \tag{5}$$

where $\lambda$ was chosen based on a 2-Fold cross-validation, with $\lambda$ in $[10e^{-7}, 10e^3]$ with log-scale spacing. Since both $Y_{:,i}$ and our embedding $O$ represent object features by positive values, we restricted $w \geq 0$. Intuitively, this means that we try to explain every SPoSE dimension by combination of the *presence* of certain affordance dimensions, not by trading them off. Figure 1 shows the predictions $\tilde{Y}_{:,i} = Ow_i^*$,

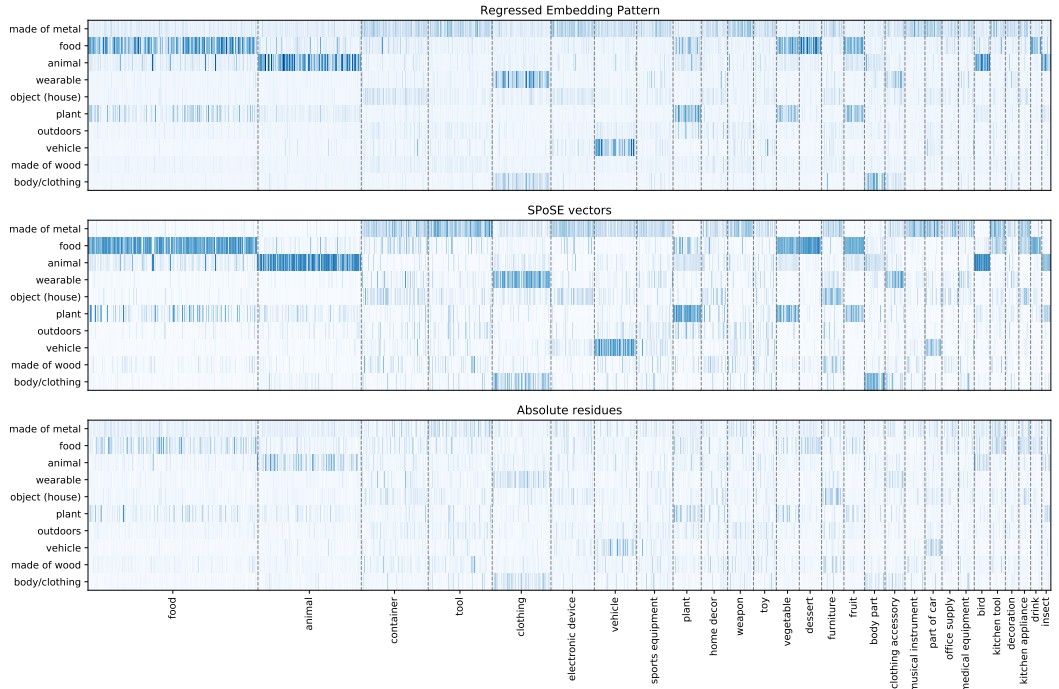

Figure 1: Prediction of first 10 SPoSE embeddings from affordance embeddings (top) versus actual SPoSE embeddings (middle). Objects are grouped by semantic category (those with $\geq 15$ objects). The absolute residuals of the prediction are also shown (bottom). Color range is fixed across plots to show the magnitude of residuals. For full diagram with all dimensions, see Figure 7 in Appendix C.

true values $Y_{:,i}$ and the absolute residues $|\tilde{Y}_{:,i} - Y_{:,i}|$ for each SPoSE dimension $i$. For clarity, objects are grouped by their semantic category and only plotted for categories with size $\geq 15$. The visual resemblance of the patterns, and the range of correlations between true and predicted dimensions (top: 0.84, mean: 0.46), see Figure 2b for distribution), indicate that the affordance embedding contains enough information to predict most SPoSE dimensions.

**Relationship between SPoSE and affordance dimensions** We first considered the question of whether affordance dimensions correspond directly to SPoSE dimensions, by looking for the best match for each of the latter in terms of correlation. As shown in Figure 2(a), most SPoSE dimensions have one or more affordance dimensions that are similar to them. However, when we consider the cross-validated regression models to predict SPoSE dimensions from affordance dimensions, *every* model places non-zero weight on several of the latter. Furthermore, those predictions are much more similar to SPoSE dimensions than almost any individual affordance dimension. Figure 2(b) plots, for every SPoSE dimension, the correlation with its closest match (x-axis) versus the correlation with the cross-validated prediction of the regression model (y-axis). The best predicted dimensions are categorical, e.g. "animal", "plant", or "tool", or functional, e.g. "can tie" or "flammable". Structural dimensions are also predictable, e.g. "made of metal", "made of wood", or "paper", but appearance-related dimensions, e.g. "colorful pattern", "craft", or "degree of red", less so.

What can explain this pattern of predictability? Most SPoSE dimensions can be expressed as a linear combination of affordance dimensions, where *both* the dimensions and regression weights are *non-negative*. This leads to a sparse regression model – since dependent variables cannot be subtracted to improve the fit – where, on average, 5 affordance dimensions have 80% of the regression weight. Each affordance dimension, in turn, corresponds to a ranking over verbs. Figure 3a shows the top 10 verbs in the 5 most important affordance dimensions for predicting the "animal" SPoSE dimension. As each affordance dimension loads on verbs that correspond to coherent modes of interaction (e.g. observation, killing, husbandry), the model is not only predictive but also interpretable. Whereas we could also use dense embeddings to predict SPoSE dimensions, they do not work as well (in either

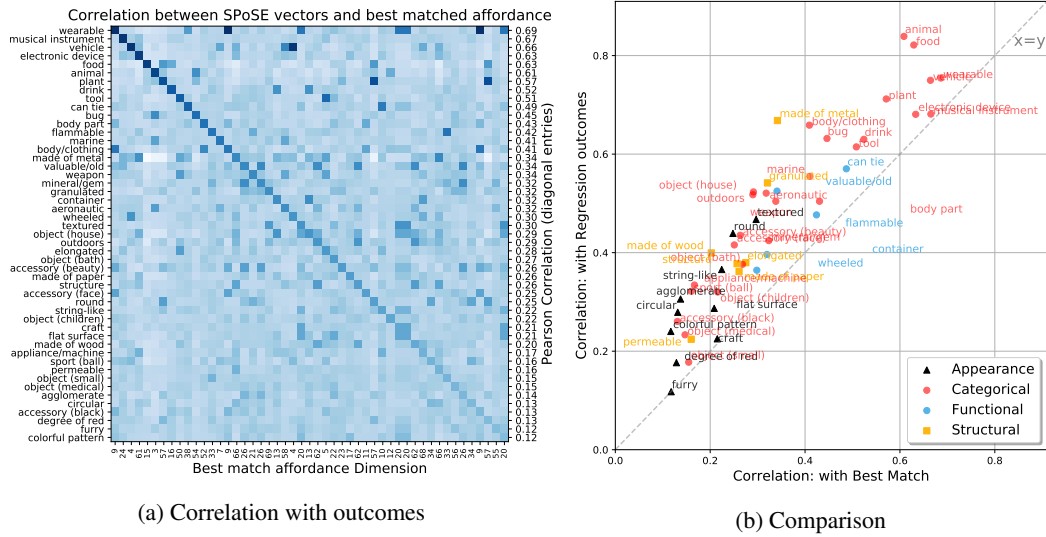

(a) Correlation with outcomes

(b) Comparison

Figure 2: **a)** Correlation between each SPoSE dimension and the corresponding best match in our affordance embedding (correlation values shown in right vertical axis). **b)** For each SPoSE dimension, the relationship between the similarity with the best matching affordance dimension (x-axis) and the similarity with the cross-validated prediction of the regression model for it (y-axis).

accuracy or interpretability, see Figure 3b for GloVe). If we consider the top 5 verbs from affordance

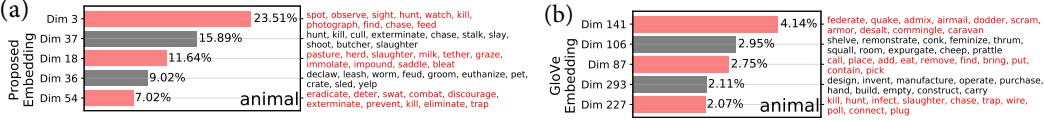

Figure 3: Top 10 verbs in the 5 most important affordance dimensions (proposed affordance embedding versus GloVe 840B) for predicting the "animal" SPoSE feature.

dimensions that are most used in predicting each SPoSE dimension, we see that "tool" has "sharpen, blunt, wield, plunge, thrust" (D50); "animal" has "spot, observe, sight, hunt, watch" (D3), "hunt, kill, cull, exterminate, chase" (D37), or "pasture, herd, slaughter, milk, tether" (D18); "food" has "serve, eat, cook, prepare, order" (D15), or "bake, leaven, ice, eat, serve" (D64); "plant" shares D2 with "food", but also has "cultivate, grow, plant, prune, propagate" (D57). The full list of affordance dimensions most relevant for predicting each SPoSE dimension is in Appendix F (Figure 8–13).

These results suggest that SPoSE dimensions are predictable *insofar* as they can be expressed as combinations of modes of interaction with objects. Using the approach in Section 3.3, we produced a combined ranking over verbs for each SPoSE dimension. We replaced the embedding $O$ in (4) with the SPoSE prediction $\widetilde{Y}$ and we ranked the verbs for dimension $h$ according to $S(\widetilde{Y}_{:,h}, \tilde{P}_k)$. Table 2 shows, for every SPoSE dimension, ranked by predictability, the top 10 verbs in its ranking. Testing against a null hypothesis of $0$ correlation, the p-values obtained range from 0.0 to the maximum of 7.61e-7 for the SPoSE dimension "furry", indicating the statistical significance of correlation with our regression outcomes. Examining this table provides empirical confirmation that, first and foremost, highly predictable categorical dimensions correspond to very clean affordances. The same is true for functional dimensions, e.g. "can tie" or "container" or "flammable"; even though they are not "classic" categories, subjects group items belonging to them based on their being suitable for a purpose (e.g. "fasten", "fill", or "burn"). But why would this hold for structural dimensions? One possible reason is that objects having that dimension overlap substantially with a known category (e.g. "made of metal" and "tool"). Another is that the structure drives manual or mechanical affordance (e.g. "elongated" or "granulated"). Finally, what are the affordances for appearance dimensions that can be predicted? Primarily, actions on items in categories that share that appearance, e.g. "textured" is shared by fabric items, "round" is shared by many fruits or vegetables. Prediction is worse when

the items sharing the dimension come from many different semantic categories ((Hebart et al., 2019) lists the pictures of items shown to subjects).

| Pearson correlation | Dimension label | Taxonomy | Affordances (Top Ten Ranked Verbs) |
|---|---|---|---|
| 0.84 | animal | categorical | kill, spot, hunt, observe, chase, feed, slaughter, sight, trap, find |
| 0.82 | food | categorical | serve, eat, cook, prepare, taste, consume, add, mix, stir, order |
| 0.75 | wearable | categorical | wear, don, match, knit, sew, fasten, rip, embroider, tear, model |
| 0.71 | plant | categorical | grow, cultivate, plant, add, eat, chop, gather, cut, dry, prune |
| 0.67 | made of metal | structural | fit, invent, manufacture, incorporate, design, position, attach, utilize, carry, install |
| 0.61 | tool | categorical | wield, grab, hold, carry, sharpen, swing, hand, pick, clutch, throw |
| 0.57 | can tie | functional | fasten, tighten, unfasten, undo, attach, thread, tie, secure, loosen, loose |
| 0.54 | granulated | structural | contain, mix, scatter, add, gather, remove, sprinkle, dry, deposit, shovel |
| 0.48 | flammable | functional | light, extinguish, ignite, throw, carry, flash, kindle, place, manufacture, douse |
| 0.47 | textured | appearance | remove, place, hang, tear, stain, spread, weave, clean, drape, wrap |
| 0.44 | round | appearance | grow, cultivate, pick, add, slice, place, eat, chop, throw, plant |
| 0.40 | made of wood | structural | place, remove, carry, incorporate, design, contain, bring, construct, manufacture, find |
| 0.40 | container | functional | empty, fill, carry, place, clean, load, bring, dump, unload, leave |
| 0.38 | elongated | structural | grab, carry, wield, hold, pick, place, throw, hand, bring, drop |
| 0.24 | colorful pattern | appearance | manufacture, buy, design, place, remove, sell, invent, purchase, contain, bring |
| 0.23 | craft | appearance | place, bring, remove, design, hang, call, buy, put, pull, manufacture |
| 0.22 | permeable | structural | fit, incorporate, remove, place, design, manufacture, install, position, clean, attach |
| 0.18 | degree of red | appearance | place, call, add, contain, remove, find, buy, bring, introduce, sell |

Table 2: Affordance assignment for a selection of SPoSE dimensions mentioned in the text, ordered by how well they can be predicted from the affordance embedding. Dimension labels are simplified. The full table, $p$-values and descriptions for each dimension are in Appendices D and E, respectively.

## 5 CONCLUSIONS

In this paper, we introduced an approach to embed objects in a space where every dimension corresponds to a pattern of verb applicability to those objects. We view such a pattern as a very broad extension of the classical notion of "affordance", obtained by considering verbs that go well beyond motor actions, and objects that encompass many different categories beyond tools or household objects. We showed that this embedding can be learned from a text corpus and used to rank verbs by how applicable they would be to a given object. We evaluated this prediction against two separate human judgment ground truth datasets, and verified that our method outperforms general-purpose word embeddings trained on much larger text corpora. This gave us with confidence that the patterns of applicability we identified captured information relevant to human judgments of affordance.

We used our embedding to predict SPoSE dimensions for objects. SPoSE is an embedding that was derived from human behavioural judgements of object similarity, and has interpretable dimensions. We used the resulting prediction models to probe the relationship between the dimensions of our embedding and the various types of SPoSE dimensions. This allowed us to conclude that our "affordance" embedding knowledge predicts 1) category information, 2) purpose, and 3) some structural aspects of the object. SPoSE dimensions to do with visual appearance were poorly predicted. Our embedding is thus sufficient for predicting most SPoSE dimensions. To be able to go further this, and conclude that our embedding is a valid model of the mental representations of objects – insofar as our uses for them go – would require additional experiments. One possibility would be to run an to explicitly ask human subjects "given objects that load highly on this embedding dimension, what can you do with them", and consider the typicality of verb answers against the weight given to those verbs by the dimension. Given that our embedding is based on language data about which verbs apply to which objects, we would expect these experiments to give verb loadings coherent with ours.

To increase prediction quality in future work, one approach will be to enrich and refine the co-occurrence matrix in larger corpora, now that the basic approach has been shown to be feasible. Another interesting future direction will be to understand how affordance could be driven by more fine-grained visual appearance properties, by considering other semantic dependencies, or jointly using text and image features such as (Wang & Tarr, 2020).

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

# A    Appendix: Hyper-parameter Selection for Non-negative Matrix Factorization

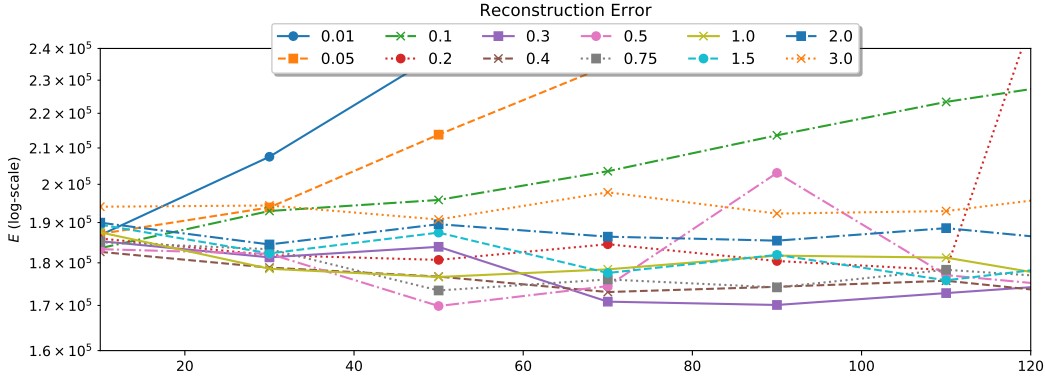

Figure 4: A zoom-in plot for the reconstruction errors.

Denote $M_t, M_v \in \{0,1\}^{n \times m}$ to be the mask matrices for indicating held-in and held-out entries of the input PPMI matrix $P$ in CV procedure, we then optimize for $O^*$ and $V^*$:

$$O^*, V^* = \underset{O,V}{\mathrm{argmin}} \|M_t \odot (P - OV^T)\|_F^2 + \beta \mathcal{R}(O, V). \tag{6}$$

To apply the multiplicative update scheme as proposed in (Lee & Seung, 2001), we first consider the partial derivatives with respect to $O$ and $V$. Denote $F(O,V) \equiv \|M_t \odot (P - OV^T)\|_F^2 + \beta \mathcal{R}(O, V)$, we have

$$\nabla_O F(O,V) = (M_t \odot OV^T)V - (M \odot P)V + \beta \cdot \mathbf{1}$$
$$\nabla_V F(O,V) = (M_t \odot OV^T)^T U - (M \odot P)^T U + \beta \cdot \mathbf{1}. \tag{7}$$

We then have the following update rules that is guaranteed to be non-increasing:

$$O^{(i+1)} \leftarrow O^{(i)} \odot \frac{(M_t \odot P)V^{(i)}}{(M_t \odot O^{(i)}(V^{(i)})^T)V^{(i)} + \beta}$$
$$V^{(i+1)} \leftarrow V^{(i)} \odot \frac{(M_t \odot P)^T U^{(i)}}{(M_t \odot O^{(i)}(V^{(i)})^T)^T U^{(i)} + \beta}, \tag{8}$$

where the fraction here represents elementary-wise division. For the choice of $M_t$ and $M_v$, we follow the same approach as proposed in (Kanagal & Sindhwani, 2010). We first split the matrix into $K$ blocks, where the rows and columns are randomly shuffled. Denote $\mathbf{r}^{(k)}$ and $\mathbf{c}^{(k)}$ to be the index vectors for rows and columns respectively, where $\mathbf{r}_i^{(k)} = 1$ if row $i$ contains in block $k$, or $\mathbf{c}_j^{(k)} = 1$ if column $j$ contains in block $k$. The mask for $k$-th block can then be expressed as $M^{(k)} = \mathbf{r}^{(k)} \otimes \mathbf{c}^{(k)}$. We then randomly select $q$ out of $K$ blocks as holdout blocks, which gives

$$M_v = \sum_{s=1}^{q} \mathbf{r}^{(k_s)} \otimes \mathbf{c}^{(k_s)}, \quad M_t = \mathbf{1} - M_v, \tag{9}$$

where $k_s$ is the index of selected block. The reconstruction error $E$ can thus be computed:

$$E = \|M_v \odot (P - O^*(V^*)^T)\|_F^2 + \beta \mathcal{R}(O^*, V^*). \tag{10}$$

Figure 4 shows a zoom-in plot of the reconstruction error under different combinations of $d$ and $\beta$. For every $(d, \beta)$ setting, we perform multiple optimization since NMF is sensitive to initialization. We then choose $d = 70$ and $\beta = 0.3$ accordingly. Empirically, we observe that the rank selection is quite robust to over-fitting when there is a sufficient sparsity control, for instance, $\beta > 0.1$ in our dataset. We also observe that whenever $d \in [50, 150]$ and $\beta \in [0.05, 0.5]$, the results are similar.

## B  APPENDIX: DISTRIBUTION OF AAUCs ON WTACTION AND MSCOCO DATASET

The following figures show the AAUC distribution of the top 5 embeddings on the WTaction and MSCOCO dataset respectively.

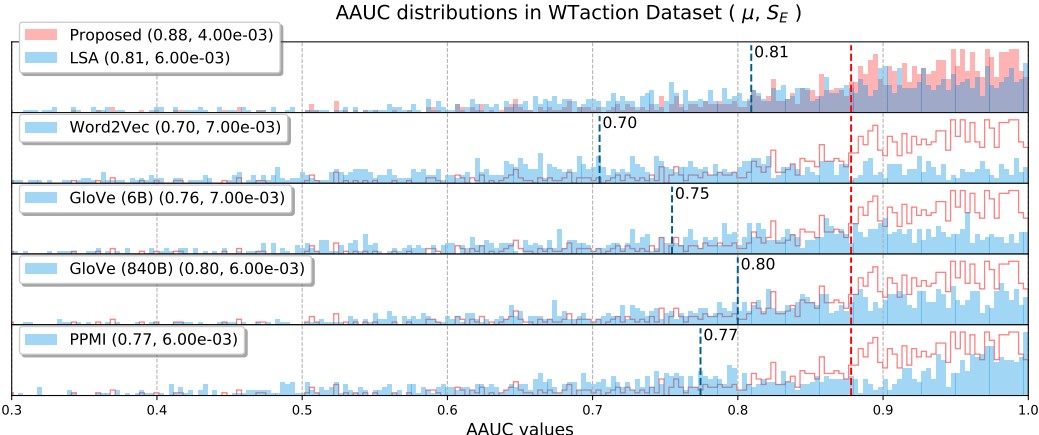

Figure 5: AAUC Distribution on WTaction Dataset

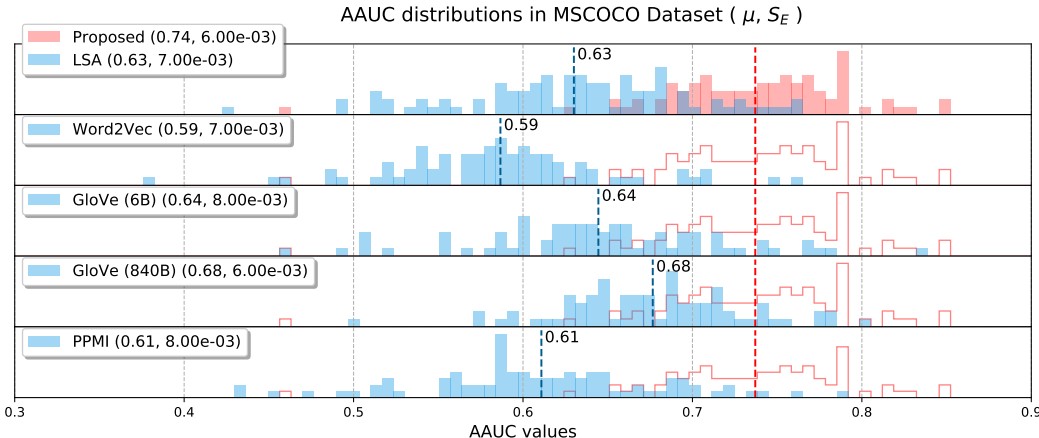

Figure 6: AAUC Distribution on MSCOCO Dataset

# C  APPENDIX: PREDICTION OF SPOSE DIMENSIONS

Figure 7: Prediction of SPoSE embeddings from affordance embeddings (top) versus actual SPoSE embeddings (middle). Objects are grouped by semantic category (those with $\geq 15$ objects). The absolute residues of the prediction is also shown (bottom). Color range is fixed to show the magnitude of residues.

# D Appendix: Affordances Assignment for each SPoSE Dimension

| Pearson correlation | p-value | Dimension label | Taxonomy | Affordances (Top Ten Ranked Verbs) |
|---|---|---|---|---|
| 0.84 | 0.0e+00 | animal | categorical | kill, spot, hunt, observe, chase, feed, slaughter, sight, trap, find |
| 0.82 | 0.0e+00 | food | categorical | serve, eat, cook, prepare, taste, consume, add, mix, stir, order |
| 0.75 | 6.9e-323 | wearable | categorical | wear, don, match, knit, sew, fasten, rip, embroider, tear, model |
| 0.75 | 9.7e-317 | vehicle | categorical | drive, hire, park, equip, rent, commandeer, crash, board, build, operate |
| 0.71 | 2.0e-271 | plant | categorical | grow, cultivate, plant, add, eat, chop, gather, cut, dry, prune |
| 0.68 | 8.6e-240 | musical instrument | categorical | hear, sound, play, learn, amplify, blare, study, tootle, toot, tinkle |
| 0.68 | 2.6e-239 | electronic device | categorical | install, operate, connect, activate, invent, disconnect, manufacture, purchase, incorporate, design |
| 0.67 | 2.5e-227 | made of metal | structural | fit, invent, manufacture, incorporate, design, position, attach, utilize, carry, install |
| 0.66 | 6.2e-219 | body/clothing | categorical | wear, don, straighten, slash, bandage, hurt, rip, injure, heal, model |
| 0.63 | 4.0e-196 | bug | categorical | kill, catch, spot, observe, find, eradicate, deter, trap, hunt, feed |
| 0.63 | 8.9e-195 | drink | categorical | drink, pour, quaff, sip, guzzle, sup, swig, spill, imbibe, gulp |
| 0.61 | 7.3e-183 | tool | categorical | wield, grab, hold, carry, sharpen, swing, hand, pick, clutch, throw |
| 0.57 | 7.3e-152 | can tie | functional | fasten, tighten, unfasten, undo, attach, thread, tie, secure, loosen, loose |
| 0.55 | 3.0e-142 | marine | categorical | spot, beach, moor, sail, observe, find, sight, capsize, catch, call |
| 0.54 | 2.6e-134 | granulated | structural | contain, mix, scatter, add, gather, remove, sprinkle, dry, deposit, shovel |
| 0.52 | 6.7e-125 | valuable/old | functional | steal, discover, find, carve, place, design, contain, recover, craft, hide |
| 0.52 | 6.0e-124 | object (house) | categorical | design, manufacture, fit, incorporate, install, place, fill, purchase, clean, buy |
| 0.52 | 1.3e-122 | aeronautic | categorical | spot, observe, sight, destroy, build, construct, find, photograph, equip, design |
| 0.52 | 7.7e-121 | outdoors | categorical | remove, construct, place, surround, incorporate, carry, erect, design, fit, build |
| 0.50 | 7.6e-114 | body part | categorical | sprain, fracture, bandage, flex, injure, rest, bruise, straighten, hurt, hyperextend |
| 0.50 | 7.9e-114 | weapon | categorical | throw, carry, hurl, drop, grab, wield, retrieve, fire, hold, toss |
| 0.48 | 3.2e-100 | flammable | functional | light, extinguish, ignite, throw, carry, flash, kindle, place, manufacture, douse |
| 0.47 | 5.5e-96 | textured | appearance | remove, place, hang, tear, stain, spread, weave, clean, drape, wrap |
| 0.44 | 1.3e-83 | round | appearance | grow, cultivate, pick, add, slice, place, eat, chop, throw, plant |
| 0.44 | 4.7e-82 | accessory (beauty) | categorical | steal, wear, find, place, gather, pick, remove, contain, give, sell |
| 0.42 | 1.1e-77 | mineral/gem | categorical | steal, discover, recover, contain, retrieve, find, hide, place, remove, incorporate |
| 0.42 | 2.8e-74 | accessory (face) | categorical | wear, remove, don, place, design, buy, call, pull, bring, find |
| 0.40 | 3.4e-68 | made of wood | structural | place, remove, carry, incorporate, design, contain, bring, construct, manufacture, find |
| 0.40 | 4.5e-67 | container | functional | empty, fill, carry, place, clean, load, bring, dump, unload, leave |
| 0.38 | 2.9e-61 | elongated | structural | grab, carry, wield, hold, pick, place, throw, hand, bring, drop |
| 0.38 | 1.4e-60 | structure | structural | incorporate, construct, fit, design, erect, position, install, mount, place, build |
| 0.38 | 3.8e-60 | object (bath) | categorical | manufacture, remove, place, invent, clean, buy, put, apply, contain, design |
| 0.37 | 1.1e-56 | string-like | appearance | remove, cut, place, pull, wrap, attach, manufacture, contain, call, bring |
| 0.36 | 3.1e-56 | wheeled | functional | drive, manufacture, hire, design, equip, fit, rent, park, purchase, invent |
| 0.36 | 2.2e-55 | made of paper | structural | manufacture, design, purchase, buy, place, invent, introduce, incorporate, fit, sell |
| 0.33 | 4.4e-47 | appliance/machine | categorical | fit, manufacture, connect, design, install, incorporate, attach, utilize, purchase, invent |
| 0.32 | 1.6e-43 | sport (ball) | categorical | manufacture, design, invent, buy, purchase, grab, carry, introduce, fit, bring |
| 0.32 | 4.3e-43 | object (children) | categorical | buy, manufacture, design, purchase, bring, find, sell, introduce, steal, call |
| 0.31 | 2.5e-39 | agglomerate | appearance | place, contain, add, remove, sell, manufacture, combine, find, buy, steal |
| 0.29 | 1.3e-34 | flat surface | appearance | place, bring, remove, put, grab, hang, wrap, manufacture, buy, pull |
| 0.28 | 9.9e-33 | circular | appearance | place, incorporate, fit, invent, remove, manufacture, design, call, position, utilize |
| 0.26 | 1.4e-28 | accessory (black) | categorical | remove, grab, manufacture, buy, place, design, bring, wear, carry, invent |
| 0.24 | 2.0e-24 | colorful pattern | appearance | manufacture, buy, design, place, remove, sell, invent, purchase, contain, bring |
| 0.23 | 4.0e-23 | object (medical) | categorical | invent, place, remove, find, contain, manufacture, design, bring, carry, buy |
| 0.23 | 1.3e-21 | craft | appearance | place, bring, remove, design, hang, call, buy, put, pull, manufacture |
| 0.22 | 1.9e-21 | permeable | structural | fit, incorporate, remove, place, design, manufacture, install, position, clean, attach |
| 0.18 | 6.5e-14 | object (small) | categorical | place, call, remove, buy, find, manufacture, introduce, contain, incorporate |
| 0.18 | 8.9e-14 | degree of red | appearance | place, call, add, contain, remove, find, buy, bring, introduce, sell |
| 0.12 | 7.6e-07 | furry | appearance | place, call, find, remove, buy, bring, introduce, contain, add, manufacture |

Table 3: Affordance assignment for SPoSE vectors ordered in terms of Pearson correlation with regression outcomes. The dimension labels are vastly simplified. The full descriptions for each dimension are provided in Appendix E (Table 4 and Table 5).

# E   APPENDIX: FULL DESCRIPTIONS FOR EACH SPoSE DIMENSION

The abbreviation and the full description of every SPoSE dimension.

| Abbreviation | Full descriptions |
|---|---|
| accessory (black) | accessories, beauty, black, blackness, classy, date, emphasize, fancy, hair, hard, high-class, manly, objects, picture, telescope |
| accessory (face) | accessories, body parts, culture, decoration, eyes, face, face accessories, facial, goes on head, hair, head, less appealing, senses, touches face, wearable |
| accessory (beauty) | accessory, beautiful, beauty, color, fancyness, feminine, feminine items, floral, flowers, flowery, gentle, girly, love, muted colors, pastel, pink |
| aeronautic | aero-nautic, air, airplanes, aviary, aviation, buoyant, flies, flight, fly, flying, flying to not, high in air, light, move, sky, swim, transportation, travel |
| agglomerate | accumulatable, bundles, collection, colors, countable, grainy to smooth, group of similar things, groupings, groups, groups of small objects, large groups, little bits of things, many, metals, nuts, objects, patterns, piles, quantity of objects in photo, round, small, small objects in groups, small parts that look alike, symmetrical |
| animal | animal, animals, animals in zoo, animals that do not fly, from complex to less, fuzzy, grass, ground animals, land animals, mammal, mammals, natural, size, wild animals, wild to human-made, wilderness |
| appliance/machine | building materials, construction, destructive, electric items, factory, farm tools, foundation, home tools, in groups, long, machinery, maintenance, mostly orange, processing, renovation, rocks, rope-like, thing, tool, tools |
| body part | body, body parts, esp extremities, extremities, extremities of body, feet, feet to hands, fingers, found on people, hand, hands, human, legs, limbs, lower body, skin |
| body/clothing | bodies, body, body accessories, body maintenance, body part- related, body parts, body parts with hair, face, how much skin showing, human, human body parts, part of body, parts, people, skin, touched by skin |
| bug | animals that stick onto things, ants, bug, bugs, can hurt you, dangerous, gardening, insects, interact with bugs, small animals, small to large, wild |
| can tie | bands, bondages, can tie, chained, circles, coils, construction, fasteners, knotting, long, rope, ropes, round, string-like, strings, tensile, thing can tie around, tied, ties, trapped, violent, wires, wrap, wrapped around to what gets wrapped |
| circular | circles, circular, cylindrical, discs, flat, round, shape, targets |
| colorful pattern | artistic, bright, bright colors, color, color variety, color vibrancy, colorful, colors, many colors, patterns |
| container | able to put something in it, boxes, buckets, can put things into, carts, container, containers, containers for liquids, containing, covering, cylinders, diverse, drums, enclosed objects, hold other things, hollow tubes, paints, shapes, storage, unknown |
| craft | a lot of patterns, art, artisinal, arts, candles, circles, color, crafts, detailed dots, do-it-yourself, grandma, grandparent-like, handmade, home patterns, home-making related, housework, in grandma's home, intricacy, quilt, rectangles, sewing, specks, stitching, twine, unknown, weaving, wood, woven, yarn |
| degree of red | color, colors, degree of redness, red, red (bright) |
| drink | 3-dimensional, beverage, containers, containers for liquids, drinks, edible, glass, glasses, hold liquid, liquid, liquids, other things, things that fit in containers, things that hold liquids, vessels with liquids, viscosity |
| electronic device | digital, digital devices, digital media, electric, electronic, electronics, hard, hard to understand, media, old technology, technological, technology, telephones, typing instruments |
| elongated | able to be held, cane, cylinders, cylindrical, darts, grouped, long, long narrow, long objects, long-shaped, narrow, pen-like, pencils, pens, shape, sharp, skinny rectangles, stick-like, sticks, straight, straight to curved, symmetrical, thin |
| flammable | fire, flammability, flammable, heat, hot, light, outdoors, warm |
| flat surface | attaching, breakable, clean, cloth, convenience, disposable, flat coverings, gathering, grated pattern, handle everyday, helpful, hold things in, multi-shaped, not smooth to smooth, paper, paper-like, sheets, stick-like, thin, things that roll, tissue, white |
| food | baked food, baked goods, carbs, cheesy, comforting, cooked, deliciousness, edible, entrees, food, made dish, natural products, nutrients, pastry, prepared food, processed, salt, where it comes from |
| furry | fluffy, furry, more of one color, white, white and fluffy, winter |
| granulated | a lot of items, ash, color, dirt, elements, grain-looking, grains, grainy, grainyness, granular objects, ground, ground (grinded), homogeneity, lots of same, many, not colorful, particles, rocky, shape, size of particles, small, small particles, stones, tiny groupings, tone, unknown minerals or drugs |
| made of metal | buckle, build, building, gray, hard, metal, metal tools, metallic, metallic tools, metals, shiny, silver, tools, use with hands |
| made of paper | books, card, classroom, collections, flatness, found in office, groups, has text, note-taking, office, paper, paper (colorful), papers, printed on, reading materials, rectangles, school, square, squares, stacks, striped, work |
| made of wood | brown, made of wood, natural, natural resources, orange, wood, wood-colored, yellow |
| marine | aquamarine life, aquatic activities, cruise, fish, in water, marine, nautical, ocean, outdoor, outside water, paradise, sea, ships, vacation, water |

Table 4: Descriptions of abbreviation (A)

| Abbreviation | Full descriptions |
| --- | --- |
| mineral/gem | beauty, clear minerals, crystal, earth-derived, gems, ice, in an artistic way, inspecting, intricate, jewelry, jewels, metallic, natural, natural minerals, prized, pure, rare, reflective, round, roundish, sharp, shiny, shinyness, sterile, translucent, valuable |
| musical instrument | control noise, hearing, instruments, listen, listening, loud, make noise, music, music instruments, musical, musical instruments, recreational instruments |
| object (bath) | bathroom, cleaning, essential everyday, gray, home (inside home), household items, hygiene, self-care, soap, toiletries, water, white |
| object (children) | baby, baby toys, child, child-like, children, dolls, toys, young, youth |
| object (house) | bland to colorful, chairs, cloth, common in household, everyday household, flat, furniture, house, house essentials, house surfaces, household commonness, household furniture, in house, living furniture, main component of room |
| object (medical) | health, health-concerning, hospital, hygiene, injury, medical, medical instruments, medical supplies, medicine, sick, to good health, unhealthy, water, wellness |
| object (small) | ?, amish, appealing, candles, circular, color? Unsure, colorful, covers, cylinders, cylindrical, flat fat cylinders, hands-on, jewelry, lids, saving, shape, similar-shaped, things you grab, twine, unknown, yellow |
| outdoors | backyard, blue, brown colored, columns, common in outdoor, dirt, garden, landscape, man-made, monuments in nature, natural, nature, not colorful, outdoor objects, outdoorsy, park, pavement, pristine, public, quiet, rocks, rough, rows, scenery, separaters, stand on their own, statues, stone, tools, wood, woodsy, yard |
| permeable | can pass through, dot patterned, grates, holed, knit, little holes, mesh, metal, net, nets, octagonal, pattern, patterned, patterns, patterns (holes), repeated patterns, repeating, repeating patterns, repetitive, shiny, silver, small, strainer |
| plant | green, green leaves, green plants, green plants and herbs, greenery, greenness, greens, grow, natural, nature, plant-like, plants, things that grow from earth, vegetable, vegetables |
| round | artistic, ball, balls, circular, circular and colorful, circus, contrasting circles, fruits, kid, pictures, round, rund, shape, spherical |
| sport (ball) | athletic, ball toys, balls) to less active, competition, recreation, round, sport, sports, sports (active, sporty |
| string-like | amount of rigid ends, confetti, different shapes, elongated, hay, high-density, knots, lines, lines jutting out, long, long things mashed up, look like sticks, mesh, netting, patterns, prickly, protruding, repeating in an ordered way, rope, ropes, skinny, spiky, stacks, strings, stringy, symmetrical, tangled, twirled around |
| structure | amusement, antenna, big, caged, common to humans, complete, disordered, electrical, elongated, enclosures, found outside, grand, high, high in air, industrial, ladder, large, multiple cylinders, multiple similar things, narrow, outdoor, part of circle, shapes, stacks, structural, tall, things that go up, things that hang, trapped, wires |
| textured | appealing, carpets, flat coverings, fractals, lay flat, mesh, pattern, patterned, patterns, pieces, rectangles, repeating shapes, repetitive, rugs, sheets, small repeating patterns, squares, textured |
| tool | elongated, hand tools, household, jagged, long, pointy tools, pole, saws, scrape, sharp, sharp tools, small instruments, straight, tools, use with hands, utility, wedges |
| valuable/old | English royalty, antiquity, bottles, bronze, fine things, gaudiness, gold, high-class, high-quality, history, important, jewelry, jewels, monarchy, old, ornaments, precious metals, pristine, royal, royalty, shiny, silver, trophy, valueable |
| vehicle | can be moved, car, cars, complex vehicles, construction, efficiency of transportation, fast, ground motorized vehicles, machinery, mobility, move, speed of movement, transportation, transportation vehicles, travel with, truck, vehicles, wheeled vehicles, wheels |
| weapon | black, danger, dangerous, equipment, masculine, military, negatively-associated, ornaments, risky, self-defense, somber, violence, violent, war, war-like, weaponry, weapons |
| wearable | accessories, blue, can wear or carry or put on, clothes, clothing, cotton clothing, covering body, shirt, things to wear, things you wear, touch body, touch person, utilities, warm clothes, wearable |
| wheeled | able to hold, bicycle, caged, can sit on, can stand or drive, carrier, chair, destinations, holds objects, light movement, mobility, motion, move someone, playful, round, thing, things with wheels, trapped, wagon, wheel, wheeled-structures, wheels |

Table 5: Descriptions of abbreviation (B)

# F APPENDIX: COMPONENTS OF SPoSE DIMENSION APPROXIMATION

The following consecutive Figures 8–13 show the component information of each SPoSE dimension. The percentage is calculated based on the portion $w_i \cdot \|O_{:,i}\|$, where $w_i$ is the regression coefficient corresponding to our $i$-th embedding dimension. The right ticks show the dimension affordances.

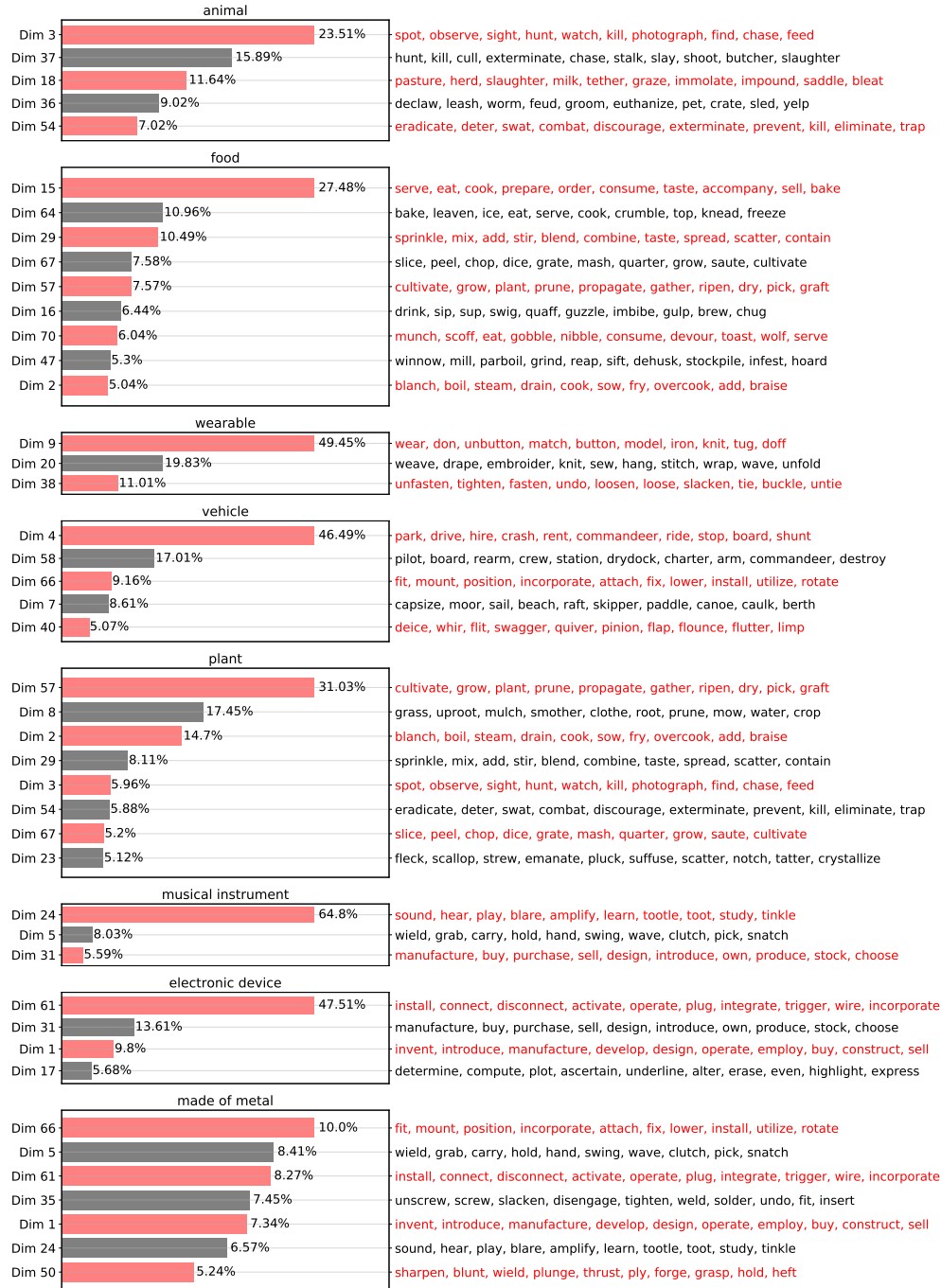

Figure 8: Components of SPoSE dimension approximation (A)

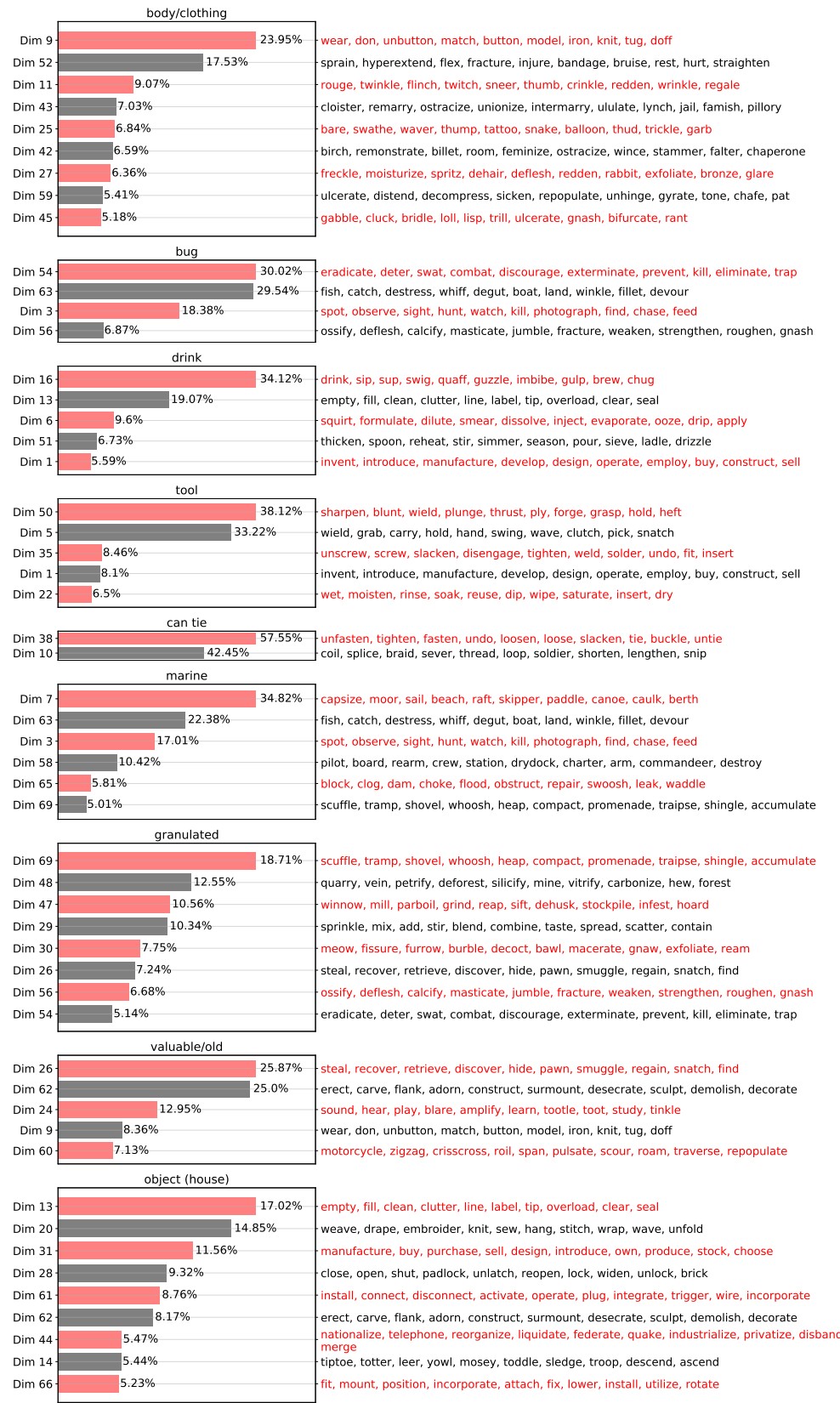

Figure 9: Components of SPoSE dimension approximation (B)

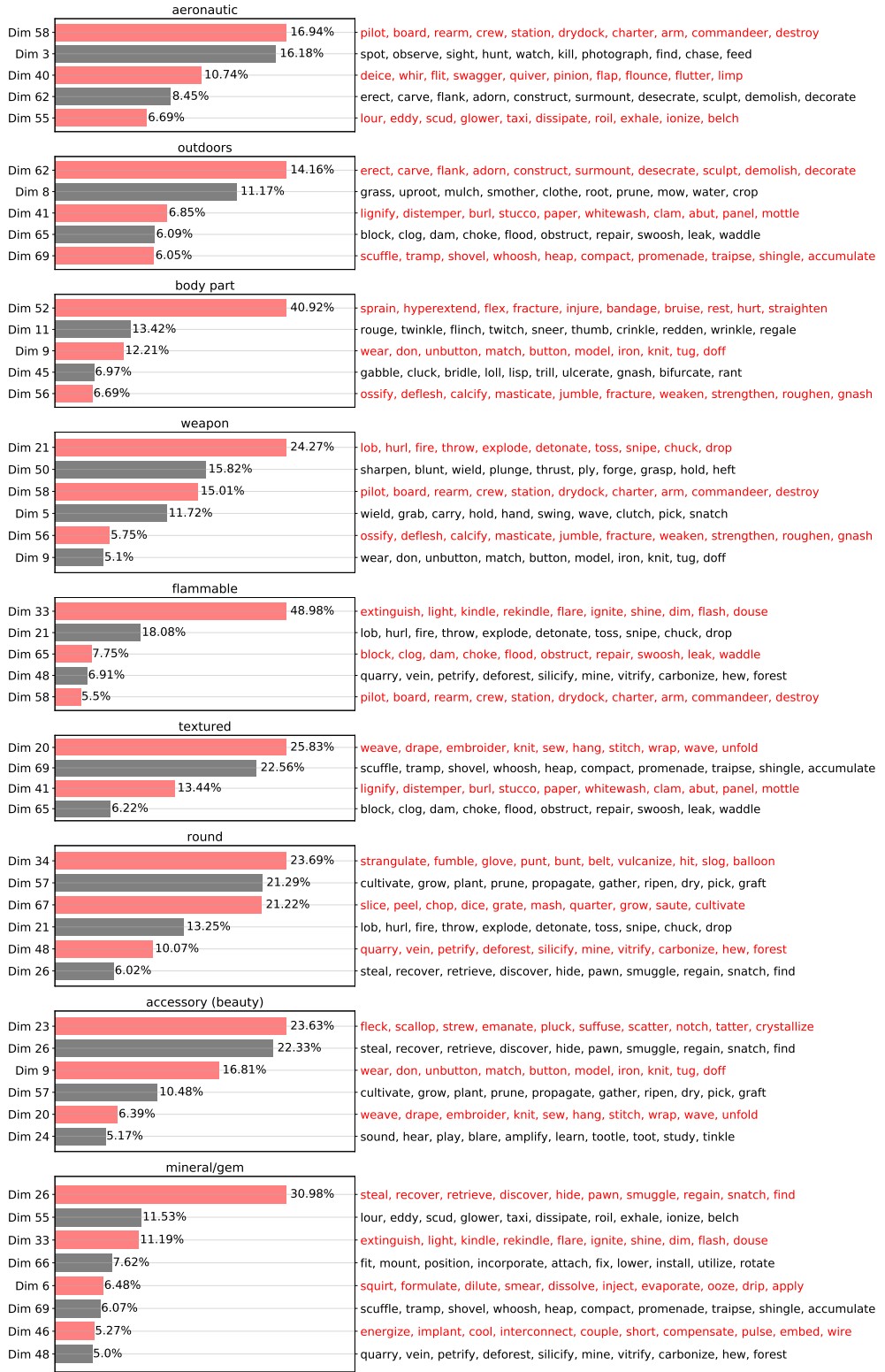

Figure 10: Components of SPoSE dimension approximation (C)

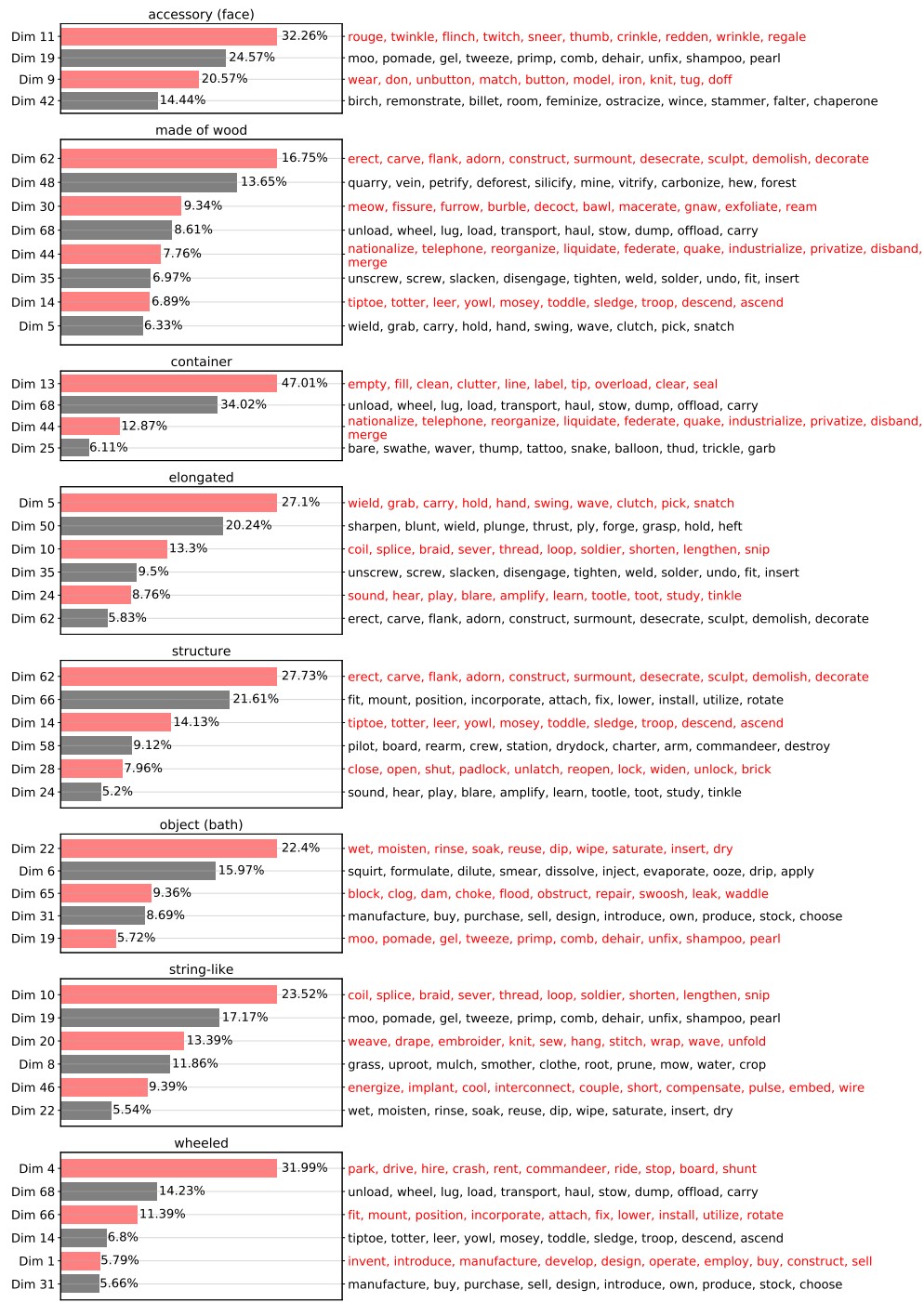

Figure 11: Components of SPoSE dimension approximation (D)

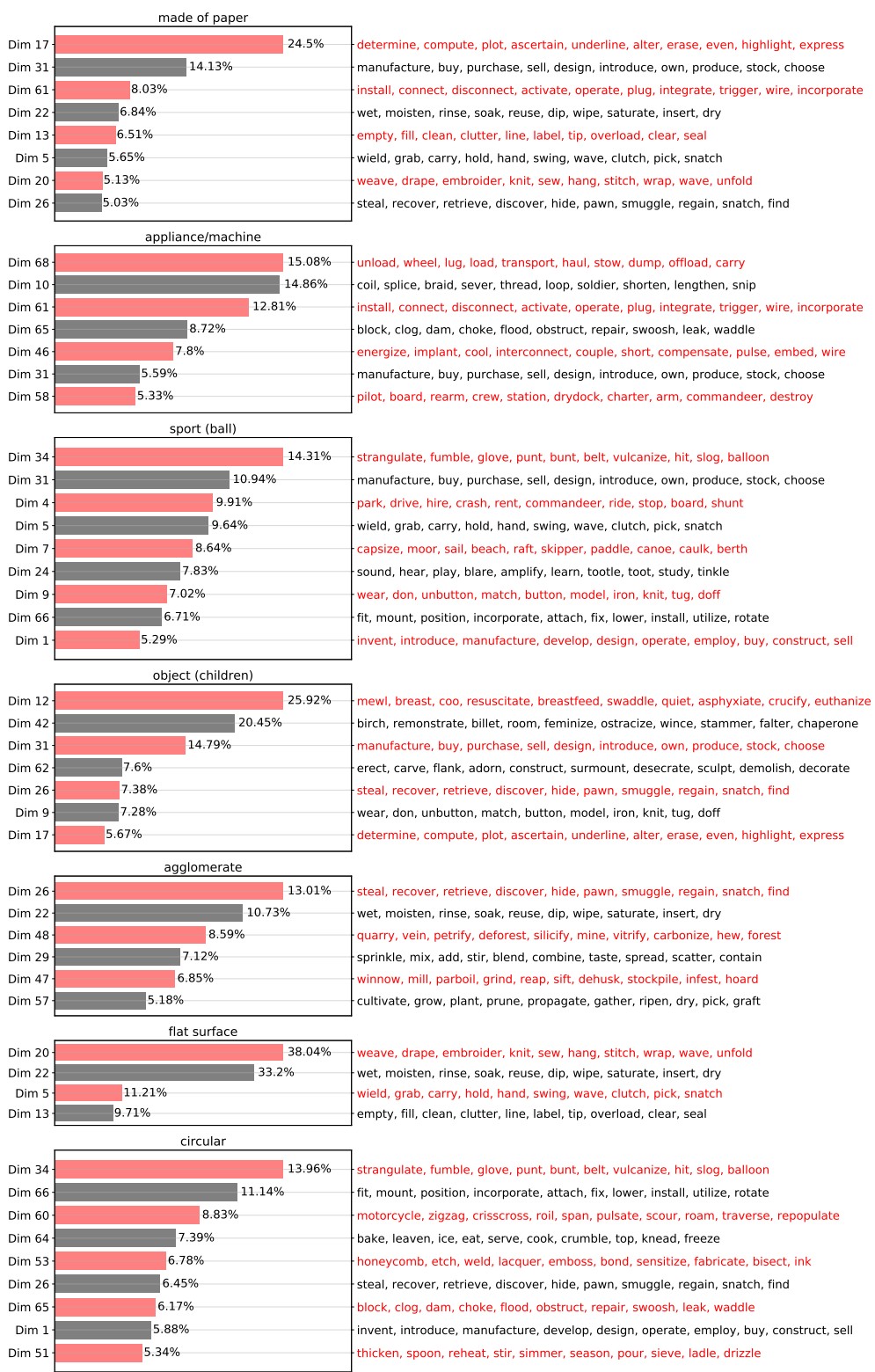

Figure 12: Components of SPoSE dimension approximation (E)

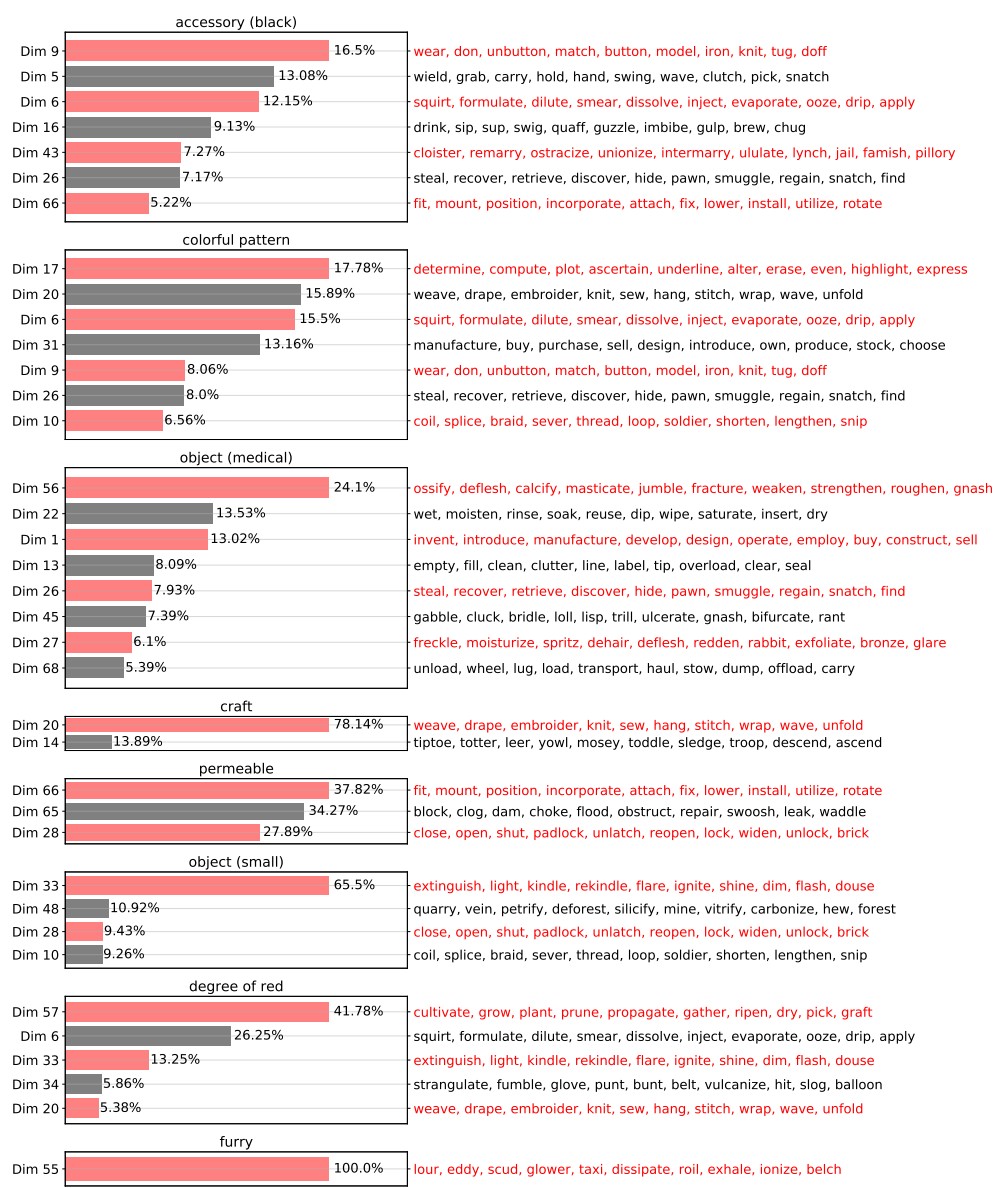

Figure 13: Components of SPoSE dimension approximation (F)

