# OpenReview forum: "Understanding Mental Representations Of Objects Through Verbs Applied To Them"
_ICLR.cc/2021/Conference — Reject_

### Official Review · AnonReviewer2 · 2020-10-26
**Review for UNDERSTANDING MENTAL REPRESENTATIONS OF OBJECTS THROUGH VERBS APPLIED TO THEM**

**Rating:** 5
**Confidence:** 4

**Review:**

General comments
---

This paper uses a factorisation of a verb-object co-occurrence count matrix to predict which verbs are applicable to which objects. This idea is related to the classical notion of an "affordance" from Gibson. The method is evaluated on a number of recent affordance datasets, obtaining better performance than a number of baseline systems.

The main problem I have with the paper as it stands is that it's not clear what the overall goal of the work is. A minor problem is that the method used appears to be entirely standard, so it's unclear what the technical contribution is. A further minor problem is that there is a whole related sub-field of computational linguistics which has been investigating a similar problem for decades which is ignored in the discussion.

The main problem: is the goal to develop a psychologically plausible cognitive model? Are we doing cogsci here? Or is it to build a knowledge base that can be used by an AI system (so more on the engineering side)? But if the latter, how would the knowledge be used, and by what sort of AI system? Is the knowledge to be used by a text-based system (if so, how) or by a situated agent interacting with an environment (in which case it needs explaining how the knowledge could be grounded in the agent's environment)?

The minor problem is that factorisation of the co-occurrence matrix appears just as standard as the other methods compared against, eg in sec. 4.1. So why are the other techniques any more baselines than yours?

The further minor problem is that the sub-field of acquiring selectional preferences in computational linguistics looks to be solving the same problem as what you have here. Classic references are Wilks from the 1970s and Resnik from the 1990s.

More specifically, there's a lot of existing work on taking a set of verb-object pairs and clustering the data in some way. This paper from 2010 is a good one to look at, and has lots of relevant references:

Latent variable models of selectional preference
Diarmuid O Seaghdha
ACL 2010

More specific comments
--

we show that the dimensions can be used to predict a state-of-the-art
mental representation of objects - it's not clear that the
representation itself is s-o-t-a; I suspect you mean that you obtained
s-o-t-a performance on an existing object-representation dataset.

"Gibson (2014) coined the term “affordance” to describe what the
environment" - the term was coined by Gibson was much earlier, 1979?

"We will refer to objects and the nouns naming them interchangeably." -
not sure what you mean here: is it that you either say "someone can
stroke a cat" or the verb "stroke" can apply to the noun "cat"? what's
the significance of the difference?

when describing the various datasets, eg sec. 3.1, some examples would
help.

Typos etc.
--

The labels in Figure 1 are too small to read.

to a particular ”mode of interaction” - left quotes

 defined as ”affordance mining” - left quotes

In this paper, we use the list of 1854 object concepts - not sure why
the number is in bold.

 The resulting list has 2541 verbs - not sure why
the number is in bold.

V is the verb loading for each of the d dimensions - "loading" is an
odd term to use here, maybe "weighting"?

 2)purpose

---

> ### Author Response · Authors · 2020-11-23
> **Responses to the technical novelty and the overall goal of the project (Part 1)**
>
> We  thank  the  reviewer  for  their  thoughtful  comments.   We  will  answer  all  questions  here,  anddescribe the edits that will be in the revised paper.
>
> > It’s not clear what the overall goal of the work is.  (...)  The further minor problem isthat the sub-field of acquiring selectional preferences in computational linguistics looks to besolving the same problem as what you have here.
>
> We very much appreciate the pointers into this literature, of which we were only tangentially aware via references in (Chao et al 2015). We will answer both questions together, if we may, as the answers are related.
>
> Our chief goal is indeed to do cognitive science and, specifically, understand the degree to which mental representations of objects are driven by what can be done to them. The cognitive model in this case is an embedding space for objects, where each dimension scores verbs by the degree that they are similarly applicable to objects (a broadly construed notion of "affordance"). We think this is a reasonable choice in itself, as a sparse factorial model is easy to interpret. However, it is also practical in two different ways. The first practical reason is that the embedding can be produced via a factorization of a matrix derived from a corpus, with additional desirable properties; we will discuss this while answering one of your other questions re how this differs from a standard factorization. The second practical reason is that it can be evaluated quantitatively in two tasks that are relevant for our goal: predicting a general purpose mental model, and predicting affordance judgements. Finally, having a dimensional model for objects makes it easier to design stimuli for future experiments involving object interactions and planning, a topic of interest to our collaborators.
>
> As you point out, identifying verbs that tend to be applied to the same nouns is a form of selectional preference. Our work is similar to (Erk 2007), (Padó et al 2007), (Ó Séaghdha 2010), (Van De Cruys 2014), and (Zhang et al 2020) in basing that identification on co-occurrence statistics in a corpus, rather than a structured resource such as WordNet (as in Resnik 1993) or labelled data (as in Bergsma 2008). The first of those five papers uses similarities of co-occurrence patterns for words to compute selectional preferences for semantic roles in FrameNet. The second uses the same similarity function to predict the plausibility of verb/relation/argument triples. Either of these could easy replace the similarity function with a similarity between embedding vectors derived from a corpus. The third paper is more similar to our approach, in that a regular topic model learns a weighing over a dictionary of elements for each latent variable (topic) in the model; it is more complex in various ways, e.g. it uses separate dictionaries for verbs and nouns, and each observation in the corpus is generated by two latent variables (instead of the single one in a regular topic model). The fourth paper trains a neural network to predict preference scores for combinations of verbs or objects, represented via embedding vectors. The fifth paper learns embeddings for individual words together with modifications for when the word is used in a certain relation. It scores combinations of words by similarity of the modified embedding vectors. The methods on these papers could be used to make the same predictions we are making in the affordance ranking task, where implementations are publicly available or feasible. The human labelled datasets we use are larger than those in the original evaluations, so this would be an interesting comparison. This said, making that prediction is not our main goal, as we discussed above, but rather a way of gauging whether our model is capturing the right information. Our proposed embedding space is a latent variable model for verb-noun applications. While this is also the case for the dual topic model in (Ó Séaghdha 2010), the internal representation in (Van De Cruys 2014), or the embeddings of nouns/verbs in (Zhang et al 2020), they would all require extensive modification to add sparsity assumptions -- important for interpretability -- and to produce combined verb rankings for embedding vectors. Doing this comparison is beyond the scope of this paper.
>
> We have edited the related work section to discuss these papers, covering broadly the same points we are making above

---

> > ### Author Response · Authors · 2020-11-23
> > **Responses to the technical novelty and the overall goal of the project (Part 2)**
> >
> > > When describing the various datasets, eg sec. 3.1, some examples would help.
> >
> > We have edited the paper to include the following details.
> >
> > "Object categories were normed in Amazon Mechanical Turk. The following 27 categories account for most of the objects: food, animal, clothing, tool, drink, vehicle, fruit, vegetable, body part, toy, container, bird, furniture, sports equipment, musical instrument, dessert, part of car, weapon, plant, insect, kitchen tool, office supply, clothing accessory, kitchen appliance, home decor, medical equipment, and electronic device (Miller, 1995). "
> >
> > "The VerbNet categories selected typically had $10-50$ verbs sharing thematic roles and selectional preferences (e.g. fill-9.8, amalgamate-22.2, manner-speaking-37.3, build-26.1, remove-10.1, cooking-45.3, create-26.4, destroy-44, mix-22.1, vehicle-51.4.1, dress-41.1.1). "
> >
> > > Factorisation of the co-occurrence matrix appears just as standard as the other methods compared against, eg in sec. 4.1. So why are the other techniques any more baselines than yours? The method used appears to be entirely standard, so it's unclear what the technical contribution is.
> >
> > While we agree that factorisation of a co-occurrence matrix is standard, we depart from this in a number of ways. Given that these ways are what leads to increased performance in the affordance prediction task, as well as interpretability of the embedding, we believe they are relevant and ask you to please consider them in detail.
> >
> > The first difference from prior work is that we start from a matrix with counts of applications of verbs to nouns, instead of all verb-noun co-occurrence in every sentence of the corpus. This reduces the size of the dataset used in learning an embedding, suggesting that the data are cleaner, the embedding method is more data efficient, or both.
> >
> > The second difference is the use of a sparse non-negative matrix factorization, as opposed to dense, real-valued factorizations such as SVD. This is necessary for producing embeddings that are interpretable, by virtue of each dimension not being present for most objects, and loading sparsely across verbs. Furthermore, it is not sufficient, given that changes in dimension $k$ and sparsity parameter $\beta$ can substantially alter the results.
> >
> > This brings us to the third difference from most other papers, which is providing a data-driven procedure to automatically determine the optimal embedding dimension $k$ and sparsity parameter $\beta$ (See Appendix A) for a matrix with these characteristics. This procedure is an adaptation of that in (Kanagal & Sindhwani, 2010), and is particularly well suited for the multiplicative update algorithm used in solving the optimization problem. We are aware of one other effort using a sparse decomposition, albeit with the goal of producing a general purpose embedding (Murphy et al 2012), and without automatic setting of either dimensionality or sparsity; our embedding performs better than theirs for both tasks, and we now include these results as well. In addition, the optimization problem is NP-hard, and the algorithm is only guaranteed to converge to a local minimum. The initial solution used is critical, and we had to investigate different approaches to find one that works well with this method.
> >
> > Given all of the differences above, we believe that the technical contribution is rather more than the use of a standard matrix factorization.
> >
> > > it's not clear that the representation itself is s-o-t-a; I suspect you mean that you obtained s-o-t-a performance on an existing object-representation dataset.
> >
> > Our apologies for this not being clear. The model for mental representations of objects that we use -- SPoSE -- was recently published in Nature Human Behaviour (12 October 2020). It is in that sense that we deemed it state-of-the-art. Although it has been available by request since 2019, it does not appear to have been used as a prediction target in any other paper, as far as we can tell.
> >
> > > Gibson (2014) should be Gibson (1979)
> >
> > Thank you for pointing out the mistake, it was introduced by the reference manager that we used. We will correct it in the revised manuscript.
> >
> > > Unclear wording: "We will refer to objects and the nouns naming them interchangeably."
> >
> > We have rephrased this to be "As we are not doing sense disambiguation for each noun that names an object, we will use noun or object interchangeably throughout the paper." As there are only 27 homonyms in 1854 nouns naming objects, this does not visibly affect the embedding. As detailed in a comment to reviewer AnonReviewer4, we are in the process of adding a sense disambiguation step to transform nouns/verbs in the corpus into WordNet synsets, and we plan to study the effect of doing so.

---

> > > ### Author Response · Authors · 2020-11-23
> > > **Responses to the technical novelty and the overall goal of the project (Part 3)**
> > >
> > > > Typos, unclear figures, other comments
> > >
> > > Thank you very much for pointing out all the typos in the paper, we will correct them in the modified manuscript. We also fixed Figure 1's labelling and moved the full diagram to Appendix. A trimmed version showing the 10 best predictions is replaced in the main part of the modified manuscript.
> > >
> > > With regards to the number object concepts/verbs being in bold, we simply wanted to make the magnitudes of the lists more visible, as much of the related work uses one or two orders of magnitude fewer items in their evaluation or vocabulary sets. We have removed the highlight.
> > >
> > > We used the term "loading" by analogy with factor analysis of a matrix dataset. Here, the object embedding would be akin to the "factors", and the verb embedding akin to the "loadings". We have edited the text to read "and V is the verb loading for each of the d dimensions, i.e. the weighting placed on each verb."

---

### Official Review · AnonReviewer1 · 2020-10-26
**Interesting motivation, but conceptually unclear and methodologically flawed**

**Rating:** 6
**Confidence:** 4

**Review:**

The authors design a distributional word embedding method inspired by Gibsonian theories of perception. They use matrix factorization techniques to derive low-rank object representations in what they call an "affordance space," linking each object to aspects of meaning shared among different types of physical actions. They argue that the learned representations are interpretable, and that this affordance space "underlies the mental representation of objects."

I unfortunately found the paper both conceptually and methodologically flawed. These criticisms fall mainly under the "Quality" and "Significance" categories, expanded below. First, a summary in pros/cons:

Pros: Designs cognitively-motivated knowledge representations; leverages a diverse set of experiments to better understand and defend these representations.

Cons: Conceptual flaws about the content of the derived representations; evaluations are insufficient to support the claims of the paper.

Quality

This paper suffers from both conceptual and methodological issues.

1. The claimed "affordance space" is not falsifiably \*about\* affordances in any deep sense. While the original data matrix linking words and their associated attested verb combinations clearly gets at possible event--object interactions, the factorized affordance space doesn't necessarily have this property. The lower-dimensional basis may span the space according to "modes of interaction" as claimed, but equally likely may describe coherent categories of contexts/places in which the actions occur, or categories of agents which perform the action, for example.I actually see three facts reported in the paper that make me think the derived data isn't about affordances per se. First, figure 2b actually shows that some of the dimensions of the affordance space best correlated with SPoSE dimensions are object-taxonomic properties. Second, the evaluation based on the raw affordance matrix (called "PPMI" in Table 1) underperforms the full model by a substantial amount, suggesting that the factorization introduces information not captured in the actual affordance data. Third and possibly most importantly, table 2 confirms that "structural" and "appearance" features are some of the best predicted features from the affordance space.The authors may argue that the set of English verbs used in the raw matrix are not the right basis for affordance knowledge, and that the factorization leads to a better abstract/conceptual affordance knowledge representation less tied to linguistic productions. But this claim about the content of the factorized representation needs to be articulated and substantiated with tests of alternative hypotheses.As a quick analogy in case my point isn't clear: you might learn word embeddings on a Wikipedia dump by factorizing a matrix of word--Wikipedia topic co-occurrence counts. The resulting low-dimensional representations aren't \*about Wikipedia topics\* in any deep sense, no matter the factorization method --- we simply talk about them as distributional meaning representations.
2. Regarding the methodology of evaluation 2: is your aim to demonstrate the necessity and sufficiency of affordance knowledge for object feature knowledge? The evaluation demonstrates a rough sort of sufficiency, but not necessity. Demonstrating necessity would require testing alternate representations, I think, which isn't reported. What do you think about this premise/issue?
3. Evaluation 1 doesn't seem very meaningful to me. It seems self-evident that representations constructed on the basis of verb--object co-occurrence data will perform well in predicting object--action co-occurrences, and probably better than representations which are not specifically tuned exclusively for that language task. (I agree that it's nontrivial that a corpus-derived representation would suffice here, but I don't find it interesting that it outperforms other more general / less task-specific corpus-derived representations.)

Significance

The aim of this paper is not clear to me. It cannot argue for a superior system of word representation, since it does not evaluate these representations on any broad evaluation tests. It also doesn't make a convincing cognitive argument about the content of mental representations, given the conceptual and methodological issues in evaluation 2, discussed above. (A convincing cognitive argument would also need to draw on data from human behavior beyond the sort gathered on AMT, or possibly neural evidence; see Mitchell et al. (2008) as an example.)

Originality

Because I haven't closely followed the relevant literature, I can't speak to the originality of the embedding method. That being said, it doesn't seem like a substantial conceptual innovation to me. I would be more motivated to let this slide if the paper were stronger on the experimental / analytic side.

Clarity

The paper is clearly written and the authors provide plenty of supporting supplemental information. Some minor comments on this front:

* Figure 1 is not very useful, either for assessing success of the method or for understanding its shortcomings. For the latter purpose, maybe consider showing the \*residuals\* of the regression, so we can understand where affordance information performs relatively better/worse across SPoSE dimensions?
* Gibson (2014) citation should be Gibson (1979).

Mitchell, T. M., Shinkareva, S. V., Carlson, A., Chang, K.-M., Malave, V. L., Mason, R. A., & Just, M. A. (2008). Predicting human brain activity associated with the meanings of nouns. science, 320(5880), 1191--1195\.

## Post-rebuttal response

I have read the other reviews and the authors' extremely thorough responses — much appreciated! See the thread below for some brief responses to the rebuttal sections in turn.
I regret posing far too high a standard in my original review. The authors' rebuttals have helped to quiet my doubts a bit, and better understand the utility of this paper as a product for cognitive science. I have accordingly revised my judgment quite a bit upward.

---

> ### Author Response · Authors · 2020-11-24
> **addressing the concerns on the methodologies, concepts and goal of the project (Part I)**
>
> We thank the reviewer for their thoughtful comments. We have taken the liberty of reordering your points, to try to group related ones and provide more concise answers.
>
> > Gibson (2014) should be Gibson (1979)
>
> Thank you for pointing out this mistake, it was inadvertently introduced by the reference manager we used. We have corrected it in the revised manuscript.
>
> > Figure 1 is not very useful, either for assessing success of the method or for understanding its shortcomings. For the latter purpose, maybe consider showing the *residuals* of the regression?
>
> The first purpose of Figure 1 is to give readers a sense of the (sparse) distribution of SPoSE dimension loadings across different object categories, for the densest dimensions. The second purpose is to illustrate the degree to which those dimensions can be predicted well, in a held-out fashion (i.e. we plot the predictions for each item when it was in the test set). We agree with the reviewer that showing the residuals of the regression would be useful to see their magnitude in relation to the original dimensions, and have added it in the revised version of the paper. Given that we cannot fit all dimensions in the current figure, we have also added a version of it with all dimension in the appendix (Figure 7).
>
> > Because I haven't closely followed the relevant literature, I can't speak to the originality of the embedding method. That being said, it doesn't seem like a substantial conceptual innovation to me.
>
> While we agree that factorization of a co-occurrence matrix is standard, we depart from this in a number of ways. Given that these ways are what leads to increased performance in the affordance prediction task, as well as interpretability of the embedding, we believe they are relevant and ask you to please consider them in detail.
>
> The first difference from prior work is that we start from a matrix with counts of applications of verbs to nouns, instead of all verb-noun co-occurrence in every sentence of the corpus. This reduces the size of the dataset used in learning an embedding, suggesting that the data are cleaner, the embedding method is more data efficient, or both.
>
> The second difference is the use of a sparse non-negative matrix factorization, as opposed to dense, real-valued factorizations such as SVD. This is necessary for producing embeddings that are interpretable, by virtue of each dimension not being present for most objects, and loading sparsely across verbs. Furthermore, it is not sufficient, given that changes in dimension $k$ and sparsity parameter $\beta$ can substantially alter the results.
>
> This brings us to the third difference from most other papers, which is providing a data-driven procedure to automatically determine the optimal embedding dimension $k$ and sparsity parameter $\beta$ (See Appendix A) for a matrix with these characteristics. We discuss this in more detail in the response to Reviewer 2.
>
> Given all of the differences above, we believe that the technical contribution is rather more than the use of a standard matrix factorization.

---

> > ### Author Response · Authors · 2020-11-24
> > **addressing the concerns on the methodologies, concepts and goal of the project (Part II)**
> >
> > > the evaluation based on the raw affordance matrix (called "PPMI" in Table 1) underperforms the full model by a substantial amount, suggesting that the factorization introduces information not captured in the actual affordance data
> >
> > We would like to answer this point before discussing the other ones, as we believe this is a misunderstanding. The two main reasons to conduct a matrix factorization are to capture patterns where verbs are applied to the same objects, and to de-noise the data while doing so.
> >
> > Factorizations of co-occurrence matrices are used extensively in computational linguistics, and we have tried to explain how our method departs from usual practices in the answer to the previous question. The role the factorization plays is not to introduce new information, but to introduce inductive bias in trying to distinguish what is noise and what is information. Further inductive bias can come from constraints, representing a priori knowledge, hypotheses, or assumptions (e.g. a sparse, positive factorization that works as well as a dense one will be more interpretable, or better for predicting sparse targets). Figure 3 aims to provide an illustration of the effect of using sparsity on interpretability.
> >
> > With regards to noise, the Stanza dependency parser is very robust, but never 100\% accurate. Even if it were, we would be operating on nouns and verbs with multiple possible meanings; going beyond this would require word sense disambiguation (something we are implementing at present). Finally, there are unusual combinations of nouns and verbs that, being infrequent, will bias the PPMI score to be higher than it should (Bullinaria and Levy, 2012). A factorization will lessen the effect of all of these noise factors, by forcing the model to only have the capacity to represent the most stable co-occurrence patterns. This is the case for SVD as well as NMF, and we use the latter, with sparsity, to achieve additional modeling goals as described above.
> >
> > > figure 2b actually shows that some of the dimensions of the affordance space best correlated with SPoSE dimensions are object-taxonomic properties.
> >
> > This is correct. SPoSE has taxonomic dimensions that act as broad category indicators (e.g. animal, food, wearable, tool, etc), as those appear to drive many judgments of object similarity, and these often have direct counterparts in our embedding dimensions. A justification of this might be that categories have strong selectional preference effects over verbs that can be applied to them. Conversely, it is hard to imagine what commonality between verbs would pertain to objects loading on some of the more visual SPoSE dimensions (e.g. "degree of red" or "colorful pattern"). Note, also, that even those taxonomic dimensions are best predicted by a combination of our dimensions, using positive weights, rather than by individual dimensions.
> >
> > > table 2 confirms that "structural" and "appearance" features are some of the best predicted features from the affordance space.
> >
> > Please note that Table 2 is only a selection of SPoSE dimensions that we picked for illustration of the range of predictability, together with the top 10 verbs in the predicted ranking. Table 3 in the Appendix lists all dimensions, and the top 50\% are almost all taxonomic/categorical. To see why there would be a few structural or appearance ones, it helps to consider their verbs and the objects loading high on them; these are not shown here but are listed on the (Hebart et al 2020) paper describing SPoSE. For structural, "made of metal" (top verbs: fit, invent, manufacture, incorporate, design, position, attach, utilize, carry, install) tend to be mechanisms or parts, "granulated" (top verbs: contain, mix, scatter, add, gather, remove, sprinkle, dry, deposit, shovel) tend to be construction materials or ingredients. For appearance, "textured" (top verbs: remove, place, hang, tear, stain, spread, weave, clean, drape, wrap) and "round" (top verbs: grow, cultivate, pick, add, slice, place, eat, chop, throw, plant) mostly apply to clothing/fabric and fruit/vegetable items. Most appearance features that are poorly predicted do not clearly correspond to items belonging to a category, or combination of categories.

---

> > > ### Author Response · Authors · 2020-11-24
> > > **addressing the concerns on the methodologies, concepts and goal of the project (Part III)**
> > >
> > > > The claimed "affordance space" is not falsifiably *about* affordances in any deep sense. (...) The lower-dimensional basis may span the space according to "modes of interaction" as claimed, but equally likely may describe coherent categories of contexts/places in which the actions occur, or categories of agents which perform the action, for example.
> > >
> > > We appreciate the concern of the reviewer, and the detailed explanation of their reasoning. Our original motivation was to look at affordance construed in a broad sense. Starting from canonical examples about motor actions applied to very specific objects (which are of concern to people in robotics, industrial design, psychology, for instance), we wondered if the obvious regularities there extended to a broader range of objects (e.g. including animals, insects, things that are not obviously manipulated) or verbs (e.g. more abstract, or focused on purpose, intent, etc). If the goal is to study lists of thousands of objects or verbs, obtaining human judgements is prohibitive, as well as fraught with complications (e.g. is a binary yes/no task removing information? how would they calibrate a verb/object compatibility score?). Given this, it makes sense to consider applications of verbs to objects in a corpus, and a factorization to extract regularities, as we discussed above.
> > >
> > > We agree that it is subjective to describe the regularities we identified as "modes of interaction"; it is the best description we could conceive of, but not the only one. We appreciate the suggestion of alternative possibilities or confounds, for future work. Some might be testable based on other information in the dependency parses (e.g. subjects of the actions), but others might be confounded (e.g. food preparation with kitchen, hunting with outdoors, tool uses with workshop, etc). Note that the latter also appears to be the case for some SPoSE dimensions (e.g. marine with colour blue, plants with colour green). We have added your point to the discussion section, but we think ruling out the other possibilities is beyond the scope of a conference paper.
> > >
> > > > Evaluation 2 demonstrates a rough sort of sufficiency, but not necessity, of affordance knowledge for object feature knowledge.(...) Demonstrating necessity would require testing alternate representations, I think, which isn't reported.
> > >
> > > In Evaluation 2, we show that affordance embedding dimensions work surprisingly well for predicting SPoSE dimensions. This is not trivially true for three reasons. First, we predict in a cross-validation, so overfitting to independent variables would lead to poor results in the held-out data. Second, the affordance dimensions are sparse, so they have to select subsets of objects. Third, as the regression weights are positive, the dimensions have to combine and cannot be traded off against each other, as in a normal regression.
> > >
> > > As you correctly point out, this shows that affordance dimensions are sufficient to explain most SPoSE dimensions. Given that SPoSE dimensions are estimated to explain judgements of object similarity, they reflect many possible types of knowledge used in making those judgements. Hence, it is informative to consider *which* SPoSE dimensions can be predicted from affordance, and which cannot. As you correctly hypothesize, one could predict SPoSE from other types of features, e.g. sparse properties as in (Zheng et al. 2019), which introduce an early version of the representation. All that these predictions allow, given the three reasons above, is to show sufficiency for some representations but not others. For example, we could consider deriving something like our affordance embedding from object/adjective relations instead of object/verb ones. We would hypothesize this would provide more information about appearance or structural SPoSE features than the affordance embedding does (as can be seen partially in Table 2 and fully in Table 3).
> > >
> > > All of this said, it is not possible to demonstrate that there are no other representations that would work as well as ours, beyond the caveat above. However, we can definitely say that we are aware of no other linear representation where dimensions are also interpretable in terms of which verbs would apply. We believe that, with additional behavioral experiments (e.g. asking subjects to produce applicable verbs for objects that score high in each of our dimensions), we could validate our embedding as a mental representation (at least to the same degree as SPoSE), and address this in further detail in an answer below.

---

> > > > ### Author Response · Authors · 2020-11-24
> > > > **addressing the concerns on the methodologies, concepts and goal of the project (Part IV)**
> > > >
> > > > > Evaluation 1 doesn't seem very meaningful to me. It seems self-evident that representations constructed on the basis of verb--object co-occurrence data will perform well in predicting object--action co-occurrences, and probably better than representations which are not specifically tuned exclusively for that language task.
> > > >
> > > > We carried this evaluation for testing whether our embedding contained the right information, i.e. verb applicability in a broad way. The embedding for an object induces a weighting over verbs that is a combination of per-dimension weights. Given this, we think that testing whether weighting ranks the verbs in a way that is compatible with affordance judgments is a reasonable lower bound. We already discuss shortcomings of this approach at some length in the paper. We do not feel the results were self-evident a priori, which is also why we included methods that use dependency parse information (e.g. DBWE or NNSE).
> > > >
> > > > > The aim of this paper is not clear to me. It cannot argue for a superior system of word representation, since it does not evaluate these representations on any broad evaluation tests. It also doesn't make a convincing cognitive argument about the content of mental representations, given the conceptual and methodological issues in evaluation 2, discussed above. (A convincing cognitive argument would also need to draw on data from human behavior beyond the sort gathered on AMT, or possibly neural evidence; see Mitchell et al. (2008) as an example.)
> > > >
> > > > SPoSE is derived purely from behavioural data, and tested by having separate subjects predict SPoSE dimensions for novel objects, based on the dimensions for known objects. Additional tests feature predictions of certain judgements (e.g. typicality) or coherence of dimension labelling by subjects. We could certainly generate predictions about the same judgments in (Hebart et al. 2020); one could argue that, given how well we can predict the dimensions, it would be trivially easy to do so.
> > > >
> > > > For a more thorough test of our embedding, we believe we would have to carry out dimension labelling experiments with a task such as "given objects that load highly on this dimension, what can you do with them". We could certainly ask subjects to produce these judgements, and confirm that they list the same verbs that score highly for that dimension. It would have been premature to run this experiment without the results shown in this paper, which we think is one argument for publishing them. Given that our embedding is based on language data about which verbs apply to which objects, we would expect these experiments to give verb loadings coherent with ours, to the degree that spoken language agrees with that in the corpus. We have added a mention of this in the discussion, as we believe this is a worthwhile direction to pursue.
> > > >
> > > > Finally, it is conceivable to test the embedding with imaging data; this is an area we have substantial experience on. Any such test would involve creating either 1) an encoding model (analogous to (Mitchell et al. 2008)), which would predict the imaging data for novel objects, based on their embedding vector; or 2) a decoding model, which would predict the embedding vector from imaging data of an object. Either approach would allow claims about whether certain types of information are present in particular regions in the brain (see "Interpreting Encoding and Decoding Models" by Kriegeskorte and Douglas 2018). However, the same issues with necessity and sufficiency that you raised earlier permeate the interpretation of the models (see "Causal interpretation rules for encoding and decoding models in neuroimaging" by Weichwald et. al. 2015). We would argue this is less interesting than behavioral predictions, given that we already know that category-level information is very decodable (from many studies reusing Mitchell 2008, or more recently shared imaging datasets with hundreds or thousands of objects, e.g. CMU BOLD5000). Non-categorical dimensions would be more interesting, and raise issues of both where the information is represented, and when it comes into play during various tasks (it might require the use of high-resolution fMRI or MEG, see "Comparison of deep neural networks to spatio-temporal cortical dynamics of human visual object recognition reveals hierarchical correspondence" by Cichy et al. 2015). Our interest in developing interpretable representations from text corpora, and other sources, is to try to leverage as much information and constraints as possible, prior to generating hypotheses for imaging experiments.

---

### Official Review · AnonReviewer3 · 2020-10-29
**Clear exposition of a method with unclear applicability**

**Rating:** 7
**Confidence:** 3

**Review:**

Summary: The authors develop object representations based on the concept of affordances, making use of a dependency parsed text corpus and a factorization of the PPMI matrix of noun-verb pairs. They show the proposed approach is able to predict human judgements of object affordances better than distributional methods and LSA. They further show that the novel representations correlate well to a set of interpretable representations that were obtained via human judgements of object similarity.

Main contributions:
1. A method of learning an embedding of objects from an unannotated text corpus which is infused with a degree of knowledge of object affordance.
2. An analysis of the relationship between the proposed embeddings and SPoSE embeddings.

Strengths:
1.  The proposed approach is simple and does not require any form of complex annotation. This enables the consideration of a large set of verbs compared to approaches which are based on manually created datasets.
2. The analysis of interpretability is well thought out. The proposed embeddings display high predictive abilities for a majority of SPoSE dimensions, suggesting that this method might offer a good model of the mental representation of objects.

Weaknesses:
1. One crucial aspect which I feel the authors neglected to adequately address is the aim of the work, and its practical applicability. What is the significance of the proposed method, beyond its ability to predict a different set of representations? If these representations are meant to achieve a new state of the art, the evaluation is too limited and fails to include common methods in the literature, such as contextual embeddings. [This has since been addressed]
2. Given the existence of methods that make use of visual features to predict affordances, I would have liked to see some form of comparison between image- and text-based methods.

## Response to comments

I thank the authors for their comments and their revision, which have clarified the aim of this work. Having read the authors' responses to this and other reviews, I realize that in my initial assessment I had misjudged the nature of the manuscript. After careful consideration, I have therefore increased my rating.

---

> ### Author Response · Authors · 2020-11-18
> **Addressing the concerns on unclear applicability**
>
> We thank the reviewer for their thoughtful comments.
>
> > What is the significance of the proposed method, beyond its ability to predict a different set of representations?
>
> Our aim is to understand how much of the mental representation of objects is driven by what can be done with/to those objects. Most of the studies looking at affordances of objects consider a much smaller universe of verbs than we do, often because they learn a predictor on labelled object/action pairs. Assembling a dataset like this would not be feasible, given the size of the verb/object lists we consider. We are interested in seeing how many of the verbs in a typical vocabulary are actually used with objects and, further, how they group together based on that usage (rather than existing classification schemes such as those in VerbNet or WordNet). The first contribution is showing that one can do this via an embedding based on dependency relations and additional constraints, where each dimension loads on related verbs much more than on all others. The task of ranking of verbs by how well they applied to objects was meant as a test of the quality of the embedding, rather than an end in itself. In doing this, we show that it is not necessary to have a labelled dataset to achieve this goal.
>
> The practical applicability stems from the second contribution, i.e. showing that the SPoSE representation can be predicted from our embedding. If we take SPoSE to be a good proxy for mental representations of objects, each SPoSE dimension that can be predicted well from our embedding can, therefore, be explained in terms of typical actions applied to the objects that have it. This carries over to the predictions of human subject judgments made from SPoSE, e.g. typicality, semantic features. A different kind of application, beyond SPoSE predictions, would be to use the embedding to define a stimulus space in experiments about object interaction (e.g. which dimensions are object specific, versus which define continua along which objects lie). This was one of our original motivations, but is not covered in this paper.
>
> > I would have liked to see some form of comparison between image- and text-based methods.
>
> We agree with the reviewer that this would be desirable, as several SPoSE dimensions are visual in nature (e.g. "textured", "round", or "colorful pattern"), while possibly also having semantic content. It would be feasible to predict them from the internal representations of deep neural networks (e.g. VGG-S or AlexNet), taking as input images of the objects in our list . We decided not do it in this paper, given limitations of space and desire to focus on actions rather than other types of information present in SPoSE. Several of the related papers we cite focus on predicting object affordances from visual features extracted from images. Given that they typically use a limited range of verbs/objects, and this prediction is not our primary goal, we opted to compare our embedding only with those derived from text.

---

### Official Review · AnonReviewer4 · 2020-10-31
**An interesting problem, but limited novelty (and missed similar, related work) for a focused problem. However, qualitative and quantitative results look good and if this could be proven useful for more than just the verb-object prediction task evaluated in the paper, this could be a good contribution!**

**Rating:** 7
**Confidence:** 4

**Review:**

Summary:

This paper attempts to learn embeddings for objects based on their affordances i.e., verbs that could be applied to them to realise their meaning. Here each dimension corresponds to an affordance or an aspect of meaning shared by actions, thus allowing a correspondence between nouns (objects) and verbs (their affordances) based on co-occurrences in text corpora in which they exist. Empirical results show that these embeddings allowing prediction of a “mental representation” of objects (i.e., in comparison to human-given annotations of dimension “semantics” in embeddings) and a qualitative analysis attempts to show how interpretable the objects are.

Reason for score:

This paper approaches an interesting problem, but is not well-placed in literature and has missed previous work that attempts to do almost the same thing; however from a different angle. I thought the question and problem was interesting enough, but given the missed references and existing embedding-learning work, there is limited novelty in this approach. However, I think the results are interesting enough and the authors did a really nice job of qualitatively analysing the representations (and additionally, it would be good to see further discussion of use-cases of this e.g., instead of just a verb-ranking task), so my score is fairly positive overall.

Positive points + questions:

1. This paper is well written and the methodology is clearly explained.

2. The empirical results show that verb rankings obtained by these embeddings outperform all previous embeddings (however those were not learned with this objective in mind, but just tested on the verb-ranking task).

3. The results on the SPoSE task (predicting which dimensions correspond to which affordances) shows that the correlations between true and predicted dimensions are high.

4. The qualitative figures and examples are very insightful and highlight te promise of this object-verb objective to learn embeddings that highlight affordances and useful semantic properties of objects.

5. On that note, it would be really interesting to see if this helps downstream tasks e.g., not just an object-verb ranking task, but a more general task (however one that does require reasoning about objects and verbs together in order to correctly solve some decision making/classification problem)


Negative points + questions:

1. Missing references: https://roboticsconference.org/program/papers/80/

2. The approach of attributing/grouping together verbs for specific dimensions seems less intuitive (and more restrained) than just allowing the embedding space to be learned by trying to make verb-object embedding representations more similar based on objective optimised for similarity/distance between embeddings. It is also further restrained given that novel verbs (unseen during training) may not be accounted for.

3. How reliable is the processing of the corpora (e.g., tokenising, bigrams, using Stanza, dependency parsing); were these sanity-checked to assess if they were correct and the amount of noise that exists? Previous work that has attempted to do this from e.g., CommonCrawl/Wikipedia data found a range of errors/inconsistencies because of the domain shift and differences in the style/type of language commonly found on the internet, thus resulting in object-verb pairs that were too noisy for predicting/training purposes. It would be helpful to see examples + a manually annotated portion for correctness.

4. The correlation results don’t have statistical significance tests/metrics and that would be helpful to see.

5. It would be interesting to see/discuss if the bigrams cause any change in performance (e.g., maybe only allowing unigrams changes the distribution of words and therefore object, verb pairs that might affect results slightly).

6. More importantly, given how prevalent subword embeddings are now, it would be interesting to see if that could be incorporated here---for e.g., if a byte pair encoding algorithm was first run over the corpus and nouns/verbs were split according to those, does this still hold? This seems important given that subword embeddings are now used (and perform better) for most of the best performing models, so if a method that allows better object-verb disambiguation could be used to kickstart tasks that require subword embeddings, that would be helpful to see! This seems very doable within this framework with a few minor changes..


Additional minor comments:

1. It is interesting that the object/verbs mined from datasets have the number of verbs nearly twice those of the objects (previous work/datasets seem to have a smaller number of verbs given the overlap of similar verbs for the same object). This begs the question of whether or not similar verbs can be collapsed into one another/and also a qualitative analysis of whether these are grouped together and can be predicted alongside.

2. It would be good to see statistical significance tests for metrics.

---

> ### Author Response · Authors · 2020-11-17
> **Addressing the concerns and elaborations on the project goal**
>
> Thank you very much for your valuable suggestions and comments. Below we will address them in details.
> > Missing references: https://roboticsconference.org/program/papers/80/
>
> Thank you for pointing us to the paper.  We will add the reference with discussion in the Related Work section, together with other work focused on identifying visual features related to object affordances. Our goal is rather different, in that we treat object affordance prediction mainly as a test of whether our embedding contains relevant information, rather than an end in itself. Furthermore, we consider thousands of verbs, rather than 50.
>
>
> > The approach of attributing/grouping together verbs for specific dimensions seems less intuitive ...
>
> Our goal was to make the dimensions be interpretable to cognitive scientists, in addition to being informative. Loadings over verbs allow for ranking over the entire verb vocabulary, and our groups of 10 (an arbitrary cutoff) are merely for illustration of what the dimension might correspond to. We agree that novel verbs or objects would not be accounted for, in principle. In practice, almost all the common verbs or objects in our list are present in this corpus, and we plan on expanding to corpora large enough (e.g. Common Crawl) that even very rare ones would be amply represented.
>
> While we considered using an supervised approach to learning an embedding based on an objective function, we decided against it given the difficulty of obtaining labelled data. As we consider thousands of objects and verbs, extracting judgements for all combinations would be prohibitively expensive, or require additional heuristics for deciding how to sample the space.
>
> > How reliable is the processing of the corpora? ...
>
> This was done in an ad-hoc fashion, by sampling about a hundred sentences. Our qualitative impression is that we are are extracting less information than we could, as the semantic parses are more likely to miss an application of a verb to a noun than to deem it present by mistake. We also rely on the lemmatization in Stanza, given that our verb lists operate on the infinitives. The other possible issue is semantic ambiguity (e.g. "grab the bat") caused by the presence of object homonyms. That said, only 27 objects out of 1854 are such that they have a homonym (e.g. "bat":animal and "bat":object are two of those 27). We are in the process of implementing word-sense disambiguation using the Lesk algorithm, so we can have object and verb lists defined in terms of WordNet synsets.
>
> > The correlation results don’t have statistical significance tests/metrics and that would be helpful to see.
>
> We have added the $p$-values for the correlation values with SPoSE (testing against a null hypothesis of 0 correlation) in the modified version of the manuscript. The $p$-values range from 0.0 to the maximum of 7.61$\mathrm{e}{-7}$ for the SPoSE dimension ``furry'', indicating the statistical significance of correlation with our regression outcomes.
>
> > It would be interesting to see/discuss if the bigrams cause any change in performance.
>
> We started this project using unigrams, and that would force us to drop 324 objects in our list. Given that we have an interest in predicting SPoSE dimensions, this was ultimately unacceptable. Subjectively, we think that having the additional objects named with bigrams may have improved the quality of the affordance dimension verb rankings. This could be because there was $>10\%$ more data, or because many spurious co-occurrences were removed (e.g. "ice_cream melts"  vs. "cream melts").
>
> > it would be interesting to see if subword embeddings, such as  byte pair encoding, could be incorporated here...
>
> Our approach could easily extend to the sub-word scenario, which would be applied on top of the Stanza output as suggested. The co-occurrence matrix would be enlarged, with each row and column corresponding to a sub-word from object list and verb list respectively. However, we believe that a sub-word embedding may not be optimal in this specific task, since some rows or columns of the co-occurrence matrix will be very sparse and therefore lead to rare co-occurrence events. This would make the matrix factorization (or viewing it as a denoising procedure) challenging, as discussed in (Turney & Pantel, 2010).

---

> > ### Author Response · Authors · 2020-11-17
> > **Addressing the concerns and elaborations on the project goal (Part 2)**
> >
> > > It is interesting that the object/verbs mined from datasets have the number of verbs nearly twice those of the objects...
> >
> > We thank the reviewer for this suggestion. We could certainly aggregate together verbs which are specializations of an action (e.g. push/shove/nudge) or,  verb synonyms (given reliable word sense labelling). This is an experiment we will try once we have evaluated our sense labelling pipeline.
> >
> > If we are considering grouping verbs after the embedding was learned, we ran a separate experiment looking at the top 50 ranked verbs associated with each well-predicted SPoSE dimension. These verbs fall naturally into $5-8$ VerbNet classes (out of $200+$ possible ones), which group together verbs based on their syntactic form and the arguments informing their semantics. This suggests that those SPoSE dimensions can be explained by very few modes of interaction (if one interprets a VerbNet class as such). However, due to page limitations, we did not report this results in our submitted version.
> >
> > > Can this approach help downstream tasks e.g., not just an object-verb ranking task, but a more general task?
> >
> > Our goal is to do cognitive science research on the mental representation of objects and, specifically, how much of that representation can be accounted for by what can be done with/to those objects. The task of ranking of verbs by how well they applied to objects was meant as a test of the quality of the embedding. More specifically, we wanted to see whether using dependency parse information would lead to better verb rankings than using a general-purpose embedding based solely on word co-occurrence. This was not a given, as datasets of the former are much sparser than the latter, for the same corpus size.
> >
> > We take SPoSE to be a a good proxy for mental representations of objects, because it can be used to make predictions of human subject judgments about those objects (of typicality, semantic features, etc). Being able to predict SPoSE means that our embedding can be used to make the same predictions; that is definitely a direction we would go in a longer study, but did not have room for here. A different kind of application would be to use the embedding to define a stimulus space in experiments about object interaction (e.g. which dimensions are object specific, versus which define continua along which objects lie).

---

### Author Response · Authors · 2020-11-23
**Manuscript Updated**

Following the suggestions and comments raised by reviewers, we have updated the submitted version accordingly.

---

### Decision · Program_Chairs · 2021-01-07
**Final Decision**

**Decision:**

Reject

**Comment:**

This paper is a computational linguistic study of the semantics that can be inferred form text corpora given parsers (which are trained on human data) are used to infer the verbs and their objects in text. The reviewers agreed that the work was well executed, and that the experiments comparing the resulting representations to human data were solid. The method employed has little or no technical novelty (in my opinion, not necessarily a flaw), and it's not clear what tasks (beyond capturing human data) representations could be applied to (again, not a problem if the goal is to develop theories of cognition).

The first draft of the work missed important connections to the computational linguistics literature, where learning about 'affordances for verbs' (referred to as 'selectional preferences') has long been an important goal. The authors did a good job of setting out these connections in the revised manuscript, which the reviewers appreciated.

The work is well executed, and should be commended for relating ideas from different sub-fields in its motivation and framing. But my sincere view is that it does not meet the same standards of machine-learning or technical novelty met by other papers at this conference. It is unclear to me what the framing in terms of 'affordance' adds to a large body of literature studying the semantics of word embeddings, given various syntactically and semantically-informed innovations.  It feels to me like this work would have been an important contribution to the literature in 2013, but given the current state of the art in representation learning from text and jointly learning from text and other modalities, I would like to have seen some attempt to incorporate these techniques and bridge the gap between the notion of affordance in text/verbs (selectional preference) and Gibson's notion of object affordance (what you can do physically with an object) in experiments and modelling, not just in the discussion. Such a programme of research could yield fascinating insights into the nature of grounding, and the continuum from the concrete, which can be perceived and directly experienced, to the abstract, which must be learned from text. I encourage the authors to continue in this direction. An alternative is to consider submitting the current manuscript to venue where the primary focus is cognitive modelling, and accounting for human, behavioural data, and where there is less emphasis on the development of novel methods or models.

For these reasons, and considering the technical scope of related papers in the programme, I cannot fairly recommend acceptance in this case.